# A nanobody toolbox to investigate localisation and dynamics of *Drosophila* titins and other key sarcomeric proteins

**Vincent Loreau[1], Renate Rees[2], Eunice HoYee Chan[1], Waltraud Taxer[2], Kathrin Gregor[2], Bianka Mußil[2], Christophe Pitaval[1], Nuno Miguel Luis[1], Pierre Mangeol[1], Frank Schnorrer[1]\*, Dirk Görlich[2]\***

[1]Turing Centre for Living Systems, Aix-Marseille University, CNRS, IDBM, Marseille, France; [2]Department of Cellular Logistics, Max Planck Institute for Multidisciplinary Sciences, Göttingen, Germany

**\*For correspondence:**
frank.schnorrer@univ-amu.fr (FS);
goerlich@mpinat.mpg.de (DG)

**Competing interest:** The authors declare that no competing interests exist.

**Abstract** Measuring the positions and dynamics of proteins in intact tissues or whole animals is key to understanding protein function. However, to date, this is challenging, as the accessibility of large antibodies to dense tissues is often limited, and fluorescent proteins inserted close to a domain of interest may affect protein function. These complications apply in particular to muscle sarcomeres, arguably one of the most protein-dense assemblies in nature, which complicates studying sarcomere morphogenesis at molecular resolution. Here, we introduce a toolbox of nanobodies recognising various domains of the two *Drosophila* titin homologs, Sallimus and Projectin, as well as the key sarcomeric proteins Obscurin, α-Actinin, and Zasp52. We verified the superior labelling qualities of our nanobodies in muscle tissue as compared to antibodies. By applying our toolbox to larval muscles, we found a gigantic Sallimus isoform stretching more than 2 µm to bridge the sarcomeric I-band, while Projectin covers almost the entire myosin filaments in a polar orientation. Transgenic expression of tagged nanobodies confirmed their high affinity-binding without affecting target protein function. Finally, adding a degradation signal to anti-Sallimus nanobodies suggested that it is difficult to fully degrade Sallimus in mature sarcomeres; however, expression of these nanobodies caused developmental lethality. These results may inspire the generation of similar toolboxes for other large protein complexes in *Drosophila* or mammals.

## Editor's evaluation

In this important study, the authors have generated a large collection of nanobodies against *Drosophila* muscle components, showing their interest to define the molecular organisation of muscle sarcomeres. Moreover, they show that nanobody expression in muscles can block the normal function of those proteins. While the use of nanobodies to reveal the distribution of proteins as such is not novel, their use in a model organism is novel and their demonstration of their usefulness is compelling. Beyond *Drosophila* and muscles their work can emulate similar strategies for other tissues in other species.

## Introduction

Muscles use their sarcomeres to generate forces that power animal movements. Sarcomere morphology is remarkably conserved from fruit flies to humans: each sarcomere is bordered by two Z-discs that cross-link the plus-ends of parallel actin filaments, while their minus ends face towards the centrally located bipolar myosin filaments (*Lange et al., 2006*; *Lemke and Schnorrer, 2017*). Both

**eLife digest** Our muscles are not just for lifting weights. They also keep us alive. For example, our heartbeat is powered by the muscles in the heart wall. Just like other organs in the body, muscles are made up of cells called muscle fibres. Each muscle fibre is divided into many smaller units, or 'sarcomeres', which contain specialised proteins that pull on each other to produce muscle contractions.

Although the structure of mature muscles is rather well understood, we know much less about how muscles develop or how they are maintained throughout adult life. Understanding this is especially important in the case of the heart, because its muscle cells are not replaced throughout our lives. Instead, the heart muscle cells we are born with are maintained as we age while working continuously. This means that the proteins within the heart muscle sarcomeres are continuously under mechanical stress and may need to be repaired. How this repair might happen is not well understood.

Nanobodies are very small versions of antibodies that recognise and bind to specific protein targets. In biological research, they are used as a tool to observe proteins of interest within cells. This is done by labelling nanobodies, for example, with chemical fluorophores or fluorescent proteins; once labelled, the nanobody binds to its target protein, and scientists can monitor its location and behaviour within the cell. Cells, and even flies, can also be genetically manipulated to produce labelled nanobodies themselves, which has the advantage of visualising the dynamic behaviour of the target protein in the living cell or organism.

To better study the proteins in muscle cells, scientists from two different research groups developed a nanobody 'toolbox' that specifically targets sarcomere proteins. First, Loreau et al. made a 'library' of labelled nanobodies targeting different sarcomere proteins in *Drosophila melanogaster* fruit flies. Second, they used this library of nanobodies to locate several sarcomere proteins in the mature sarcomeres of different fly muscles. Third, using flies that had been genetically altered to produce the labelled nanobodies in their muscle cells, Loreau et al. were able to observe the behaviour of the target proteins in the living muscle. Together, these experiments showed that one protein in *Drosophila* that is similar to the human sarcomere protein titin has a similar size to the human version, whereas a second *Drosophila* titin-like protein is shorter and located at a different place in the sarcomere. Both of these proteins work together to stabilise muscle fibres, which is also the role of human titin.

The nanobodies generated here are a significant contribution to the tools available to study muscle development and maintenance. Loreau et al. hope that they will help reveal how sarcomere proteins like titin are maintained, especially in the heart, and ultimately how the heart muscle manages to continue working throughout our lives.

filaments are stably linked by the gigantic titin spring protein, which in mammals binds with its N-terminus to α-actinin at the Z-disc and is embedded with its C-terminus at the M-band of the sarcomere. Thus, titin determines the length of the mammalian sarcomere (*Linke, 2018*; *Luis and Schnorrer, 2021*; *Tskhovrebova and Trinick, 2003*).

As sarcomere architecture is well-conserved, *Drosophila* is an excellent model to study how a sarcomere is built during development (*Katzemich et al., 2013*; *Katzemich et al., 2015*; *Orfanos et al., 2015*; *Weitkunat et al., 2017*; *Weitkunat et al., 2014*). Generation of monoclonal antibodies against *Drosophila* sarcomere proteins has been helpful to locate key proteins within the mature sarcomere (*Burkart et al., 2007*; *Ferguson et al., 1994*; *Katzemich et al., 2012*; *Lakey et al., 1990*; *Qiu et al., 2005*; *Szikora et al., 2020*). However, to date, there is no comprehensive toolbox of antibodies recognising defined domains of the large sarcomeric proteins, in particular against defined domains of the two large *Drosophila* titin homologs, Sallimus (Sls) and Projectin (gene called *bent*, *bt*). Such tools would be valuable to study how the sarcomeric machine assembles during morphogenesis.

Recent gene-tagging approaches have generated a substantial number of *Drosophila* transgenic lines, each expressing one sarcomeric protein fused to green fluorescent protein (GFP) at its C-terminus (*Sarov et al., 2016*) or at a random internal position (*Buszczak et al., 2007*; *Kanca et al., 2017*; *Kelso et al., 2004*; *Nagarkar-Jaiswal et al., 2015*). Nevertheless, a number of these tagged lines label only a subset of protein isoforms or result in homozygous loss of function phenotypes as the GFP-tagged protein cannot fully support the function of the endogenous protein in the dense

sarcomeric environment (*O'Donnell et al., 1989*; *Orfanos and Sparrow, 2013*; *Orfanos et al., 2015*; *Sarov et al., 2016*). Hence, GFP tagging does not always provide an optimal solution to study the native dynamics of a sarcomeric protein.

These limitations motivated us to develop an alternative to antibodies and GFP-tagged lines for sarcomeric proteins. We chose the recent camelid nanobody technology because of the small size of the nanobodies (~4 nm, 12–15 kDa), their single-chain protein nature, and their potentially high affinity against target domains (*Muyldermans, 2013*; *Pleiner et al., 2018*; *Pleiner et al., 2015*). As nanobodies can be used on fixed samples or fused to a fluorescent protein and expressed in living tissues, nanobodies are ideal tools to quantify the position and dynamics of sarcomeric proteins in their dense environment.

Thus far, the application of nanobodies to the *Drosophila* model has been largely restricted to commercially available GFP and mCherry nanobodies that allowed localisation, trapping, or degradation of GFP-tagged or mCherry-tagged proteins in *Drosophila* tissue (*Ákos et al., 2021*; *Caussinus et al., 2011*; *Harmansa and Affolter, 2018*; *Harmansa et al., 2017*; *Harmansa et al., 2015*). Recently, nanobodies were used to locate *Drosophila* proteins tagged with short artificial nanotags (*Vigano et al., 2021*; *Xu et al., 2022*). However, nanobodies that recognise endogenous *Drosophila* protein domains have not been reported.

Here we generated a nanobody toolbox against seven different epitopes of the two *Drosophila* titin homologs, Sallimus (Sls) and Projectin (Proj). Additionally, we raised nanobodies against two epitopes of the core M-band protein Obscurin and against epitopes of the key Z-disc proteins α-Actinin and Zasp52. We verified their specificity as well as their superior penetration and labelling efficiencies compared to antibodies. Applying our nanobodies to *Drosophila* muscle tissues confirmed the expression of different Sls isoforms in different muscle types and identified a gigantic more than 2-μm long Sls protein in larval muscles. It further showed that Projectin is bound to the myosin filament in a strictly polar fashion, resembling the mammalian titin homolog. It also revealed unexpected differences in Obscurin isoform expression in different muscle types. Finally, by generating transgenic animals that express nanobody fusions to NeonGreen or degradation signals, we established that nanobodies are suitable tools to monitor the dynamics or manipulate the function of endogenous sarcomeric proteins in intact living animals.

## Results

### *Drosophila* anti-sarcomere immunogen design

Mammalian sarcomere length is determined by a long titin protein isoform that spans linearly from the Z-disc to the M-band and thus adopts a length of about 1.5 μm in relaxed human muscle (*Brynnel et al., 2018*; *Fürst et al., 1988*; *Linke, 2018*). To investigate to what extent the localisation of this critical component of bilaterian muscle is conserved across evolution, we aimed to reinvestigate the localisation of the two *Drosophila* titin homologs, Sallimus and Projectin by generating nanobodies specific for them. By carefully mining the Flybase expression database (http://flybase.org/reports/FBgn0086906; http://flybase.org/reports/FBgn0005666), we have annotated the likely longest Sallimus (Sls) and Projectin (Proj, gene called *bent*, *bt*) isoforms expressed in larval body wall muscles (*Figure 1A and B*). The longest Sls isoform contains 48 immunoglobulin (Ig) domains of the total 52 Ig domains coded in the Sls gene (four are selectively present in a short larval isoform). Additionally, Sls contains long stretches of flexible regions rich in amino acids proline, valine, glutamic acid, and lysine (PEVK) that form an elastic spring as well as five C-terminal fibronectin (Fn) domains. In flight muscles, these long PEVK stretches are, however, not expressed resulting in a short indirect flight muscle Sls isoform (*Figure 1A*; *Burkart et al., 2007*; *Spletter et al., 2018*). This domain organisation largely resembles the I-band part of mammalian titin (*Tskhovrebova and Trinick, 2003*).

Conversely, the long Projectin isoform contains 35 Ig and 39 Fn domains that are organised mainly in Ig-Fn super-repeats with a consensus myosin light chain kinase domain close to its C-terminus (*Figure 1B*; *Ayme-Southgate et al., 2008*). This domain organisation largely resembles the A-band part of mammalian titin (*Tskhovrebova and Trinick, 2003*).

To generate nanobodies against Sls and Projectin domains, we selected a subset of small domains that, according to published transcriptomics data (*Spletter et al., 2015*; *Spletter et al., 2018*), should be expressed in most muscle types, such as larval, flight, or leg muscles. We chose domains close to the

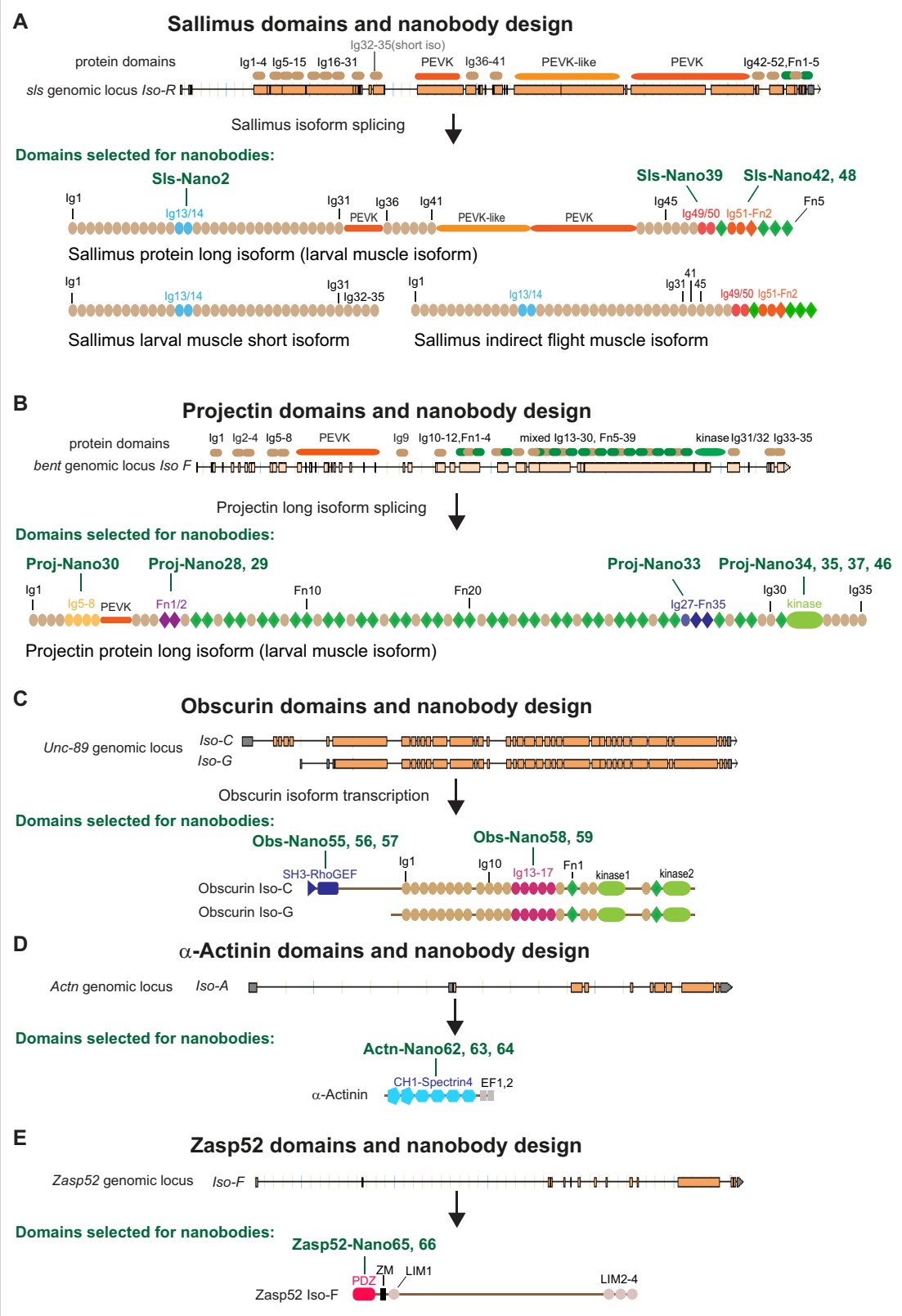

**Figure 1.** *Drosophila* nanobody design. (**A–E**) Schematic of Sallimus (**A**), Projectin (**B**), Obscurin (**C**), α-Actinin (**D**), and Zasp52 (**E**) genes and their protein domain organisation. Top: genome loci taken from Flybase with exons represented by orange boxes and introns by black lines. The coded protein domains are overlayed with Immunoglobulin (Ig) domains in brown, proline, valine, glutamic acid, and lysine (PEVK) in orange, Fibronectin (Fn) and kinase domains in green for Sls and Projectin. Bottom: predicted domain organisation of the proteins in the respective isoforms. Domains selected for nanobody production are highlighted by special colours, and the names of the respective nanobodies are indicated above the protein.

N- and C-termini of both proteins to assess their possible extended configuration in sarcomeres. We successfully expressed Sls-Ig13/14, Sls-Ig49/50, and Sls-Ig51-Fn2 recombinantly and generated the respective nanobodies Sls-Nano2, Sls-Nano39, Sls-Nano42, and Sls-Nano48 against these domains (*Figure 1A*). For Projectin, we selected Ig5-8, Fn1/2, and Ig27-Fn35 as well as the kinase domain to generate Projectin nanobodies Proj-Nano30, Proj-Nano28 and 29, Proj-Nano33, and Proj-Nano34, 35, 37, and 46 that recognise these domains, respectively (*Figure 1B*).

In a complementary approach to both titins, we also selected two regions in the core M-band protein Obscurin (gene called *Unc-89*), namely the N-terminal SH-3/RhoGEF and the middle Ig13-17, to raise the nanobodies Obs-Nano55, 56, and 57, and Obs-Nano58 and 59, respectively (*Figure 1C*). Furthermore, we expressed full-length α-Actinin to generate nanobodies against its CH1-Spectrin4 domains, named Actn-Nano62, 63, and 64, as well as nanobodies recognising the PDZ domain of the core Z-disc component Zasp52, called Zasp52-Nano65 and 66 (*Figure 1D and E*).

---

## anti *Drosophila* protein nanobody production pipeline

**1. Immunogen preparation**

a) *Drosophila* flight muscle dissection and myofibril isolation

b) recombinant antigen production

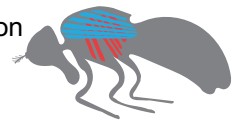

**2. Alpaca immunisation**

a) two immunisations with dissected myofibrils

b) three additional immunisations with purified proteins

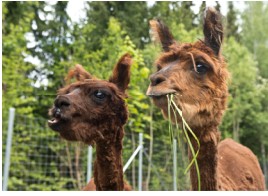

**3. Nanobody library and selection**

a) construction of immune library

b) phage display with immobilised target protein

c) sequencing and clone selection

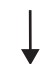

**4. Nanobody labelling and purification**

a) nanobody expression in *E. coli* and purification

b) nanobody conjugation with fluorescent dyes or biotin

c) affinity test against recombinant target protein

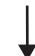

**5. Nanobody staining of *Drosophila* larval muscles**

actin Sls-Nano2,Sls-Nano39 actin, Sls-Nano2,Sls-Nano39

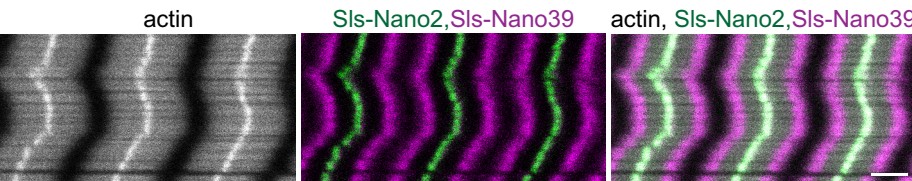

**Figure 2.** Nanobody production pipeline. Overview of our optimised nanobody production pipeline against *Drosophila* sarcomeric protein domains. See text for details. Scale bar is 3 µm.

## Anti-*Drosophila* titin nanobody generation

To produce a comprehensive set of nanobodies against the above-selected Sls, Projectin, Obscurin, α-Actinin, and Zasp52 domains, we used two sources of immunogens (*Figure 2* for the workflow). First, we hand-dissected the indirect flight muscles from 1000 wild-type adult flies and isolated their myofibrils, which express large amounts of the selected target domains (*Spletter et al., 2018*). These were used for two immunisations of a single alpaca. We rationalised that this base immunisation of the alpaca should be advantageous for future immunisations with selected sarcomeric protein domains. Second, we recombinantly expressed selected Sls and Projectin domains as His$_{14}$-SUMO or His$_{14}$-NEDD8-tagged proteins in *Escherichia coli* and purified them by binding to a Ni(II) chelate matrix, followed by extensive washing and elution with a tag-cleaving protease (*Frey and Görlich, 2014a*). These recombinant antigens (100 µg each) were used for three consecutive immunisations. Four days after the last immunisation, a blood sample was taken, lymphocytes were recovered, and total RNA was isolated and reverse-transcribed into cDNA. Finally, a phage display library with a complexity of more than $10^8$ independent clones was constructed. This followed a previously described workflow (*Pleiner et al., 2015*; *Pleiner et al., 2018*).

To isolate high-affinity nanobodies, we employed three rounds of phage display, using low concentrations (0.1–1 nM) of baits. Coding sequences of selected nanobodies were sequenced in a 96-well format and classified according to their sequence similarity. Selected clones were then expressed as His$_{14}$-SUMO-tagged or His$_{14}$-NEDD8-tagged fusions in *E. coli* and purified by the affinity-capture-protease elution strategy (*Frey and Görlich, 2014a*), with typical yields of 20–50 mg nanobody per litre bacterial culture.

## Nanobody labelling and affinities

For application in fluorescence microscopy, we labelled the nanobodies directly through one or two ectopic cysteines placed at the N- and C-termini with appropriate fluorophore maleimides (*Pleiner et al., 2015*; *Pleiner et al., 2018*). Labelling was performed 'on-column', i.e., after binding the His$_{14}$-SUMO-tagged nanobodies to Ni(II) chelate beads. Washing of the beads allowed for the convenient removal of free fluorophore before the tag-free labelled nanobodies were eluted with the tag-cleaving protease. Using this workflow, labelling was almost quantitative, as indicated by the size shifts on sodium dodecyl sulfate–polyacrylamide gel electrophoresis (SDS-PAGE) (*Figure 3A*) and by a ratiometric measurement of the optical densities at the absorption maxima of protein and the fluorophore.

To measure the binding affinity (K$_D$) of a nanobody to its target, we chose Sls-Nano2, labelled it with biotin, and immobilised it on high-precision streptavidin Octet sensors for biolayer interferometry (*Figure 3B*). On- and off-rates of the cognate Sls Ig13/14 domains were then measured by allowing a concentration series to bind and subsequently dissociate from the nanobody (*Figure 3B*). The data indicate a nearly irreversible binding with an on-rate of ~$10^6 \cdot M^{-1} \cdot s^{-1}$, an off-rate in the order of $10^{-5} \cdot M^{-1} \cdot s^{-1}$ and K$_D$ in the 10 pM range. These values are at the limit of what can be reliably measured with this technology. The high affinity can be explained by *Drosophila* proteins being highly immunogenic in mammals, by the very large immune repertoire of alpacas, and by our particularly stringent selection from a very large immune library.

## Anti-Sallimus and Projectin nanobody specificity

To assess the specificity of our nanobodies and the efficiency of labelling muscle tissue, we first assayed how well they label late-stage *Drosophila* embryonic muscles. We fixed wild-type stage 17 embryos and incubated them with fluorescently labelled Sls or Projectin nanobodies and performed confocal microscopy. Most of our nanobodies efficiently stained embryonic muscles, showing the expected striated pattern of Sls and Projectin in stage 17 embryos (*Figure 4A and B* and *Figure 4—figure supplements 1 and 2*). Thus, in total, we generated 12 different Sls and Projectin nanobodies against three different Sls and four different Projectin epitopes.

To test the specificity of the nanobodies, we generated embryos in which we depleted either the Sls or Projectin protein by muscle-specific RNAi driven by *Mef2*-GAL4 (*Schnorrer et al., 2010*), followed by a double staining with anti-Sls and anti-Projectin nanobodies. We found that in all cases the staining of Sls or Projectin was severely reduced after the knock-down of the respective protein, demonstrating the specificity of our nanobodies (*Figure 4* and *Figure 4—figure supplements 1 and 2*). In each case, we found that the striated pattern of the other protein was lost, demonstrating that

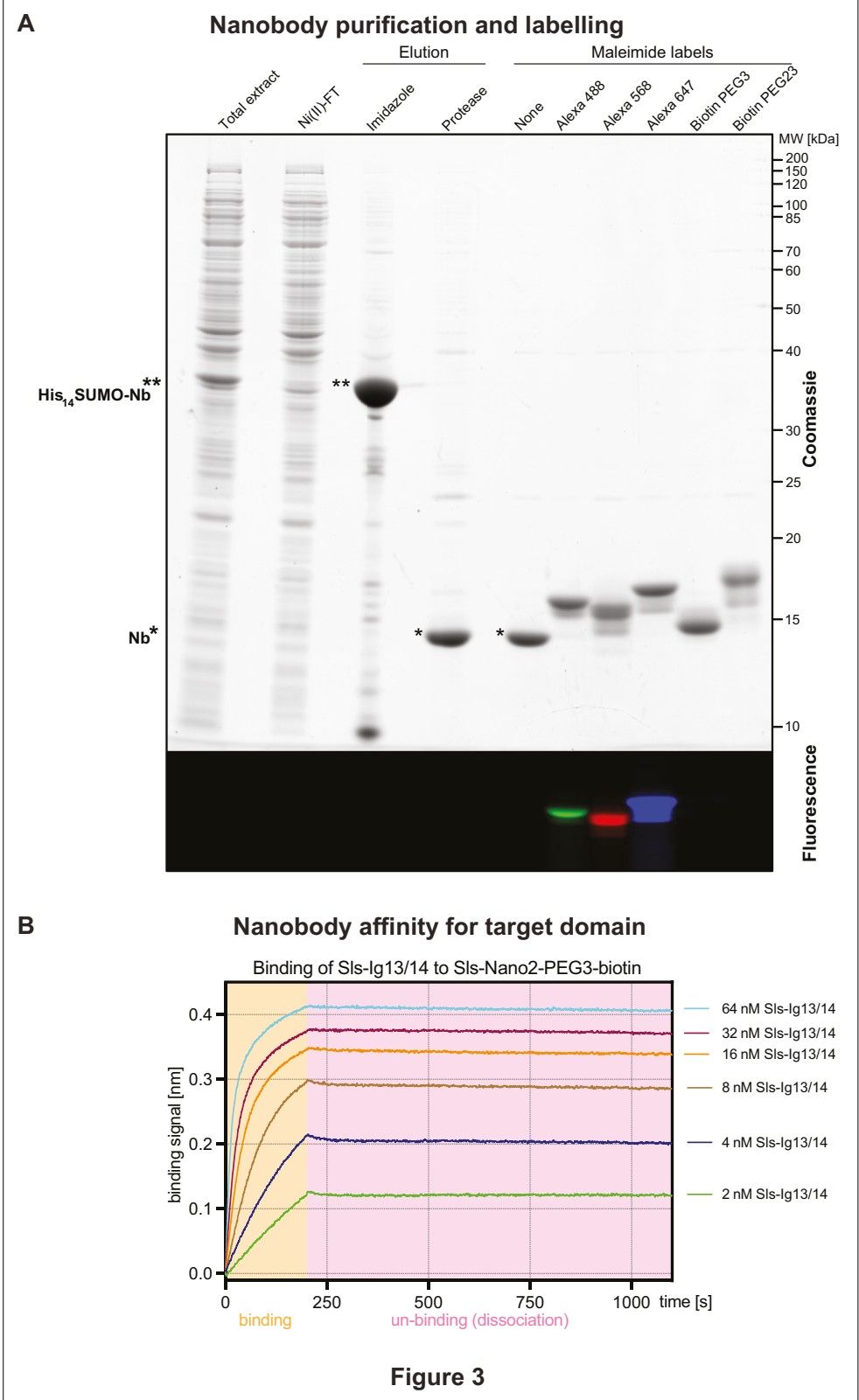

**Figure 3.** Nanobody labelling and affinity test. (**A**) Sodium dodecyl sulfate–polyacrylamide gel electrophoresis gel documenting the expression, purification, tag cleavage, and labelling of a nanobody, here Sls-Nano2. Top part stained with Coomassie blue, lower part shows fluorescence of the same gel. Note the efficiency of the labelling (essentially quantitative) by the size shift of the bands. (**B**) Nanobody affinity assay. Sls-Nano2-biotin was

*Figure 3 continued on next page*

*Figure 3 continued*

immobilised to high precision streptavidin Octet sensors to a binding signal of 0.4 nm. After washing, the target domain Sls-Ig13/14 was allowed to bind at the indicated concentration for 200 s (beige box), followed by a 900 s dissociation step (pink box). A global fit of the curves indicates a 10 pM $K_D$.

The online version of this article includes the following source data for figure 3:

**Source data 1.** SDS-PAGE of a nanobody purification and labelling example.

---

both Sls and Projectin are required to generate striated sarcomeres in stage 17 embryos. We conclude that our nanobodies specifically recognise the various Sls and Projectin domains against which they were raised and hence should be valuable tools to study the roles of the *Drosophila* titin homologs in sarcomere biology.

## Anti-Obscurin, α-Actinin, and Zasp52 nanobody specificity

*Obscurin (Unc-89)* mutants result in viable but flightless animals (*Katzemich et al., 2012*). Hence, we could test anti-Obscurin nanobody specificity in adult indirect flight muscles (called flight muscles for the remainder of the manuscript). We found that all five different nanobodies that we generated either against the N-terminal SH3-RhoGEF domains (Obs-Nano55, 56, 57) or against the central Ig13-17 domains of Obscurin (Obs-Nano58, 59) specifically label the M-band (*Figure 5A*, *Figure 5—figure supplement 1*), as had been described with established antibodies or GFP fusions (*Katzemich et al., 2015*; *Sarov et al., 2016*). This localisation is strongly reduced or absent in the hypomorphic Obscurin allele *Unc-89[EY15484]* (*Figure 5A*, *Figure 5—figure supplement 1*), demonstrating the specificity of the anti-Obscurin nanobodies.

To assay the anti α-Actinin and anti-Zasp52 nanobody specificities, we used muscle-specific RNAi. Muscle-specific RNAi of α-Actinin results in larval lethality (*Schnorrer et al., 2010*), with strongly reduced α-Actinin signal at the Z-disc, showing the specificity of our three new nanobodies (*Figure 5B*, *Figure 5—figure supplement 2A*). Similarly, we found that the Z-disc signal of both anti-Zasp52 nanobodies is specifically lost upon muscle-specific RNAi of Zasp52 (*Figure 5C*, *Figure 5—figure supplement 2B*). Knock-down of both proteins appears to affect the actin organisation at the Z-disc in different ways, as the strong phalloidin signal at the Z-disc of larval muscles is lost in *Actn-IR*, while it is broadened in *Zasp52-IR*. Taken together, we conclude that all our 22 novel nanobodies against 11 different domains result in specific detection of the target protein in muscle tissue.

## Nanobodies display superior labelling and penetration efficiencies

Nanobodies are only 13 kDa and ~4 nm in size (*Helma et al., 2015*; *Pleiner et al., 2015*), making it ideal to place a label close to the domain of interest. To illustrate another size-related advantage (*Fang et al., 2018*), we stained flight muscles with Sls-Nano2 (binding Sls-Ig13/14) and compared them to the endogenously expressed M-band protein Obscurin-GFP or to a staining with an anti-Sls antibody (anti-Kettin, binding Sls-Ig16) (*Kulke et al., 2001*). We imaged 10 μm thick z-stacks to quantify label diffusion into the thick flight muscle fibres. Because of light scattering and the fundamental limits of confocal imaging, intensities of endogenously expressed labels also reduce with imaging depth (*Sarov et al., 2016*). Using the same imaging conditions and the same fluorophore for Sls-Nano2 and the combination of anti-Sls primary and secondary antibodies, we found that the Sls-Nano2 intensity decay over z-depth is about 2.5-fold less than that of the anti-Sls antibody label (*Figure 6A–C*, *Figure 6—figure supplement 1*). This strongly suggests better penetration of the nanobody into the muscle samples compared to the larger primary and secondary antibodies. In fact, diffusion of the nanobody into the tissues appears not limiting for the image quality.

To directly compare the diffusion of the differently sized labels in the same samples, we double-stained flight muscles with Sls-Nano2 and the traditional Sls antibody. We swapped the dye colours to rule out any bias of the excitation wavelength on penetration depth. We found that Sls-Nano2 readily diffuses into the thick flight muscle samples, whereas the Sls antibody is limited to the top layer of myofibrils (*Figure 6—figure supplement 2A, B*). This demonstrates the favourable diffusion properties of the small nanobodies in the very dense and crowded environment of adult flight muscles. Labelling of myofibrils in the past was often achieved on isolated myofibrils to improve antibody accessibility (*Burkart et al., 2007*; *Szikora et al., 2020*), but myofibril isolation may change sarcomere

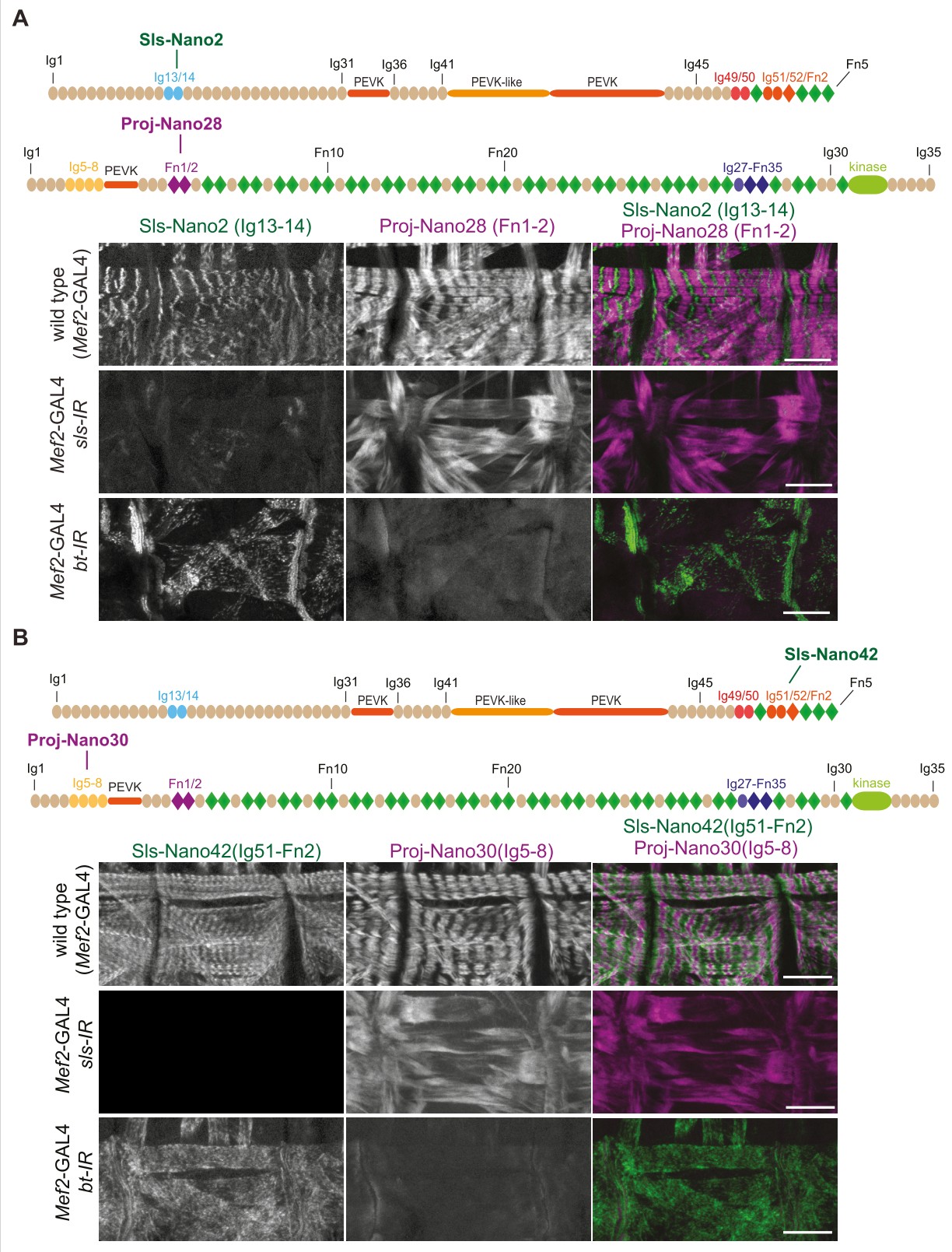

**Figure 4.** Anti-Sallimus and Projectin nanobody specificity. (**A and B**) Top: schematic representation of Sallimus or Projectin domains with nanobodies used for stainings. Bottom: scanning confocal images of stage 17 embryos of wild-type (*Mef2*-GAL4) and muscle-specific *sls* or *bt* knock-down (*Mef2*-GAL4, UAS-*sls-IR* and *Mef2*-GAL4, UAS-*bt-IR*, respectively) stained with Sls (green) and Projectin (magenta) nanobodies Sls-Nano2 and Proj-Nano28 (**A**)

*Figure 4 continued on next page*

*Figure 4 continued*

or Sls-Nano42 and Proj-Nano30 (**B**). Note the striated pattern of Sls and Projectin in wild type, which is lost upon knock-down of one component. Scale bars are 20 µm.

The online version of this article includes the following figure supplement(s) for figure 4:

**Figure supplement 1.** Anti-Sallimus and Projectin nanobody specificity.

**Figure supplement 2.** anti-Sallimus and Projectin nanobody specificity.

mechanics and thus lead to unwanted mechanical or structural artefacts (*Ayme-Southgate et al., 2004*; *Kulke et al., 2001*).

We further tested the labelling of muscles by our nanobodies in late stage 17 embryos, which have already deposited a larval cuticle (*Moussian, 2010*), impeding the penetration of large labels. In stage 16 embryos, we found the expected co-localisation of Sls-Nano2, with the anti-Sls antibody, as well as the co-localisation of Proj-Nano30 recognising Proj-Ig5-8 with an anti-Projectin antibody (*Figure 6— figure supplement 3A–C*). Both Sls and Proj proteins are not yet displaying a striated pattern as sarcomeres have not yet been assembled at stage 16. While our nanobodies stained the body muscles of stage 17 embryos well, which displayed the striated pattern of the first formed sarcomeres, neither anti-Sls (anti-Kettin), Mhc, nor Projectin antibodies produced a good staining pattern (*Figure 6— figure supplement 3A–C*). Together, we conclude that the here generated nanobody toolbox allows efficient labelling of sarcomeres in large flight muscles or whole-mount late-stage embryos.

## Sallimus and Projectin localisation in mature muscles

We next investigated adult *Drosophila* flight muscles, which show a specialised fibrillar morphology of their myofibrils and sarcomeres, caused by the expression of a specific combination of sarcomeric protein isoforms (*Schönbauer et al., 2011*; *Spletter et al., 2015*). Co-staining flight muscles with the Sls-Nano2, which recognises Sls-Ig13/14 close to the N-terminus of Sls, and the Sls-Nano42 recognising Sls-Ig51-Fn2 close to the C-terminus of Sls, revealed single and overlapping bands present at the sarcomeric Z-disc (*Figure 7A*). This pattern is expected since flight muscles contain a very short ~100 nm wide I-band (*Burkart et al., 2007*). The Sls-Nano42 band has a smaller cross-sectional radius compared to Sls-Nano2, suggesting that Sls-Ig51-Fn2 is not present in all the Sls isoforms expressed during the final stages of myofibril maturation that complete radial myofibril growth (*González-Morales et al., 2019*; *Spletter et al., 2018*). We found the same pattern for the other C-terminal Sls nanobodies, Sls-Nano39 and Sls-Nano48 (*Figure 7—figure supplement 1A*). Such central localisation of the long Sls isoforms in flight muscle sarcomeres has been reported previously with the anti-Sls antibody B2, which likely recognises Sls-Ig36-41 domains (*Burkart et al., 2007*), thus further confirming the specificity of our domain-specific Sls nanobodies.

Next, we investigated the localisation of Projectin in flight muscles and found that staining for the N-terminal portion of Ig5-8 with Proj-Nano30 resulted in a single band overlapping with the Z-disc, whereas Proj-Nano37, which recognises the Projectin kinase domain at its C-terminal end, resulted in two bands right and left of the I-band, likely overlapping with the myosin filament (*Figure 7A*). The same patterns were found with our other N- or C-terminal anti-Projectin nanobodies (*Figure 7— figure supplement 1B*). Hence, the anti-Projectin nanobodies established that Projectin is oriented linearly in flight muscles, with its N-terminus being closer to the Z-disc and its C-terminus facing the myosin filaments. Quantifying the precise positions of the Sls and Projectin domains bound by our nanobodies in flight muscles requires super-resolution microscopy, which is reported in an accompanying manuscript (*Schueder et al., 2023*).

In contrast to flight muscles, *Drosophila* leg or larval muscles have longer I-bands, likely caused by the expression of longer Sls splice isoforms that include large parts of the flexible PEVK spring domains, making these muscles softer (*Burkart et al., 2007*; *Spletter and Schnorrer, 2014*). However, the precise positions of the N- and C-terminal ends of Sls in these muscle types remained to be determined. To address this open question, we prepared fixed adult hemithoraces and L3 larval fillets and stained leg or larval body muscles with nanobodies that recognise the N- or C-terminus of Sls. The Sls-Nano2 signal overlaps with the Z-disc in leg and larval muscles, similar to flight muscles. However, Sls-Nano42, which recognises the C-terminal portion of Sls-Ig51-Fn2, showed two distinct bands with larger distances in larval muscles compared to leg muscles (*Figure 7B and C*). This demonstrates that

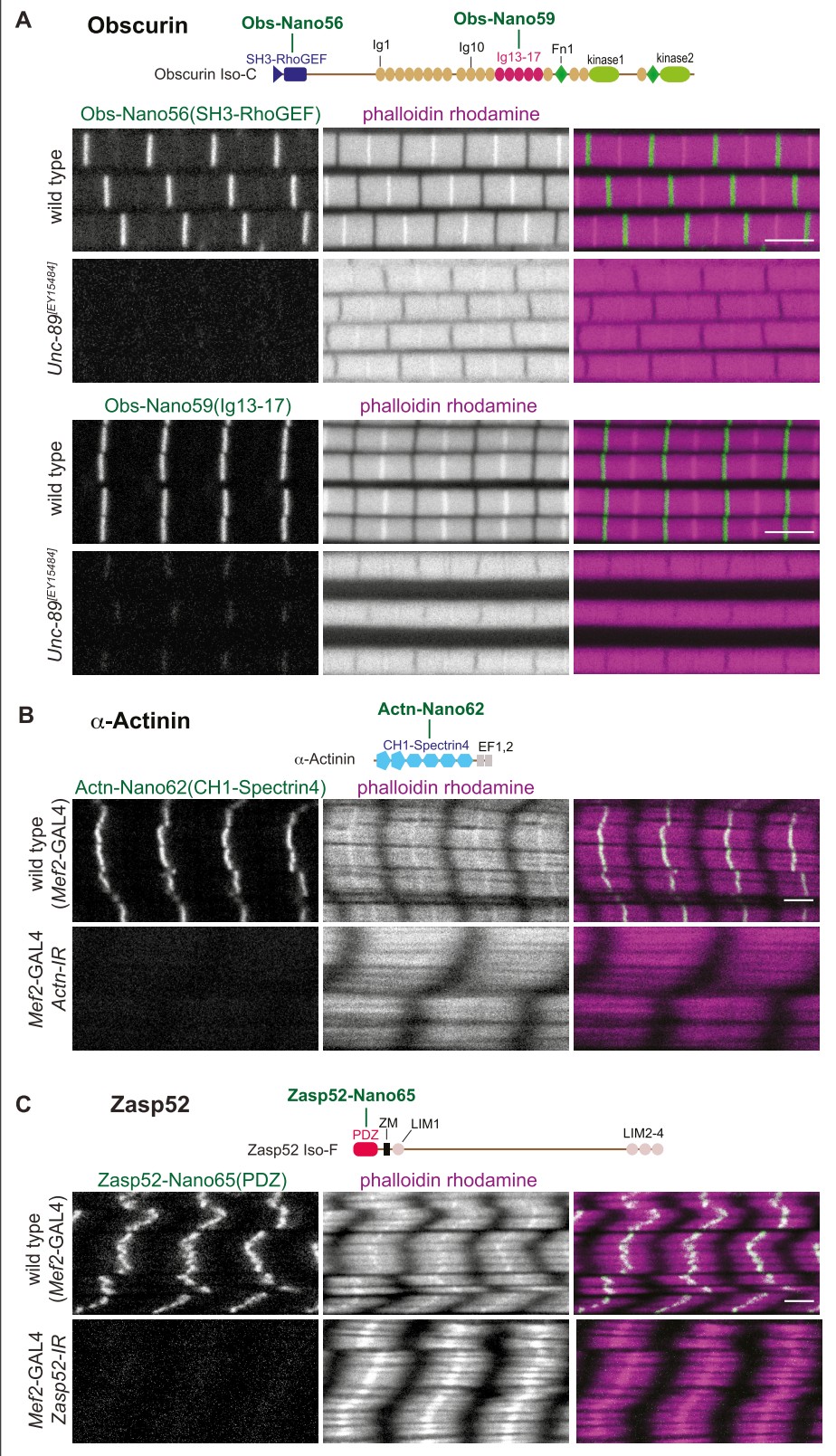

**Figure 5.** Anti-Obscurin, α-Actinin, and Zasp52 nanobody specificity. (**A – C**) Top: schematic representation of Obscurin, α-Actinin and Zasp52 domains with nanobodies used for stainings. Bottom: scanning confocal images of wild-type and *Unc-89[EY15484]* adult flight muscles stained with the indicated nanobodies and phalloidin (**A**), wild-type and *Actn* (**B**), or *Zasp52* (**C**) knock-down in larval muscles (*Mef2*-GAL4, UAS-*Actn-IR* and *Mef2*-GAL4,

*Figure 5 continued on next page*

*Figure 5 continued*

UAS-*Zasp52-IR*, respectively) stained with the indicated anti α-Actinin or Zasp52 nanobodies and phalloidin. Scale bars are 3 μm.

The online version of this article includes the following figure supplement(s) for figure 5:

**Figure supplement 1.** Anti-Obscurin nanobody specificity.

**Figure supplement 2.** Anti-α-Actinin and Zasp52 nanobody specificity.

*Drosophila* Sls is extended as a linear molecule bridging from the Z-disc likely to the myosin filament in sarcomeres with long I-bands.

In contrast to its defined location in flight muscles, earlier studies using anti-Projectin antibodies suggested that Projectin largely decorates the thick filament in *Drosophila* leg muscles (*Lakey et al., 1990*; *Saide et al., 1989*; *Vigoreaux et al., 1991*). Consistent with these reports, staining of adult leg or larval body muscles with nanobodies that recognise the N- and C-terminal portions of Projectin, Proj-Nano30, and Proj-Nano37, respectively, showed two large blocks, instead of sharp bands located on the myosin filaments in both adult leg and larval body muscles (*Figure 7B and C*). These results demonstrate that Projectin decorates the myosin filaments of cross-striated *Drosophila* muscles.

## α-Actinin, Zasp52, and Obscurin in larval muscle

As expected, we found that our nanobodies are detecting α-Actinin and Zasp52 at the Z-disc of larval muscle sarcomeres (*Figure 7D*), the well-established location for these core Z-disc components (*Jani and Schöck, 2007*; *Schnorrer et al., 2010*). More surprisingly, we found that anti-Obscurin nanobodies recognising the central Obs-Ig13-17 domains show the expected M-band pattern; however, the ones recognising the N-terminal Obs-SH3-RhoGEF domains show no staining in larval muscle (*Figure 7D*), while they do show the expected pattern in flight muscles (*Figure 5A*, *Figure 5—figure supplement 1*). This suggests that the shorter Obscurin isoform annotated in Flybase (see *Figure 1C*) is specifically expressed in larval muscles. This matches with whole larval transcriptomics data of *Obscurin* (http://flybase.org/reports/FBgn0053519). This finding further demonstrated the domain specificity of the here generated nanobodies.

## Sallimus is stretched across long I-bands

To quantify the precise length of Sls in relaxed larval muscle sarcomeres, we measured the distances between the maxima for Sls-Nano2 and Sls-Nano42 peaks. We found that Sls extends over more than 2 μm in relaxed L3 sarcomeres that are about 8.5 μm long (*Figure 8A*) and thus Sls is extended longer than the human titin protein in skeletal muscle (*Linke, 2018*). We verified the length of Sls by staining with a second Sls nanobody close to the Sls C-terminus, Sls-Nano39, that recognises Sls-Ig49/50 (*Figure 8—figure supplement 1A*). To test if the Sls C-terminus can indeed reach the beginning of the myosin filament, we co-stained larval muscles with Sls nanobodies together with an anti-Mhc antibody (*Figure 8B*). Indeed, we found that Sls-Nano42 localises Sls-Ig51-Fn2 to the beginning of the myosin filaments, demonstrating that each long Sls isoform indeed stretches across the entire long I-band of larval muscles, likely to mechanically link the Z-discs to the myosin filaments.

## Projectin is oriented on the thick filament

When carefully analysing the overlap of Proj-Nano30 and Proj-Nano37 staining, we surprisingly found that these blocks were slightly shifted with respect to each other. N-terminal Proj-Nano30 staining was located closer towards the Z-disc, whereas the C-terminal Proj-Nano37 staining was closer towards the M-band (see *Figure 7B and C*). We wanted to verify this surprising finding and double stained larval muscles with additional combinations of N- and C-terminal Projectin nanobodies, namely Proj-Nano28 that recognises Proj-Fn1/2 with Proj-Nano34 that recognises the Projectin kinase domain and Proj-Nano29 that recognises Proj-Fn1/2 combined with Proj-Nano35 also recognising the kinase domain. Again, we found that both nanobody combinations label two blocks located on the myosin filaments, with Proj-Fn1/2 located closer to the Z-disc and the Projectin kinase domain located closer to the M-band (*Figure 8—figure supplement 1B, C*). Furthermore, we obtained the same result with a fourth combination of nanobodies, Proj-Nano29, that recognises Proj-Fn1/2 and Proj-Nano33 binding to Proj-Ig27-Fn35 (*Figure 8—figure supplement 1D*). This 'shifted-blocks' pattern is not a

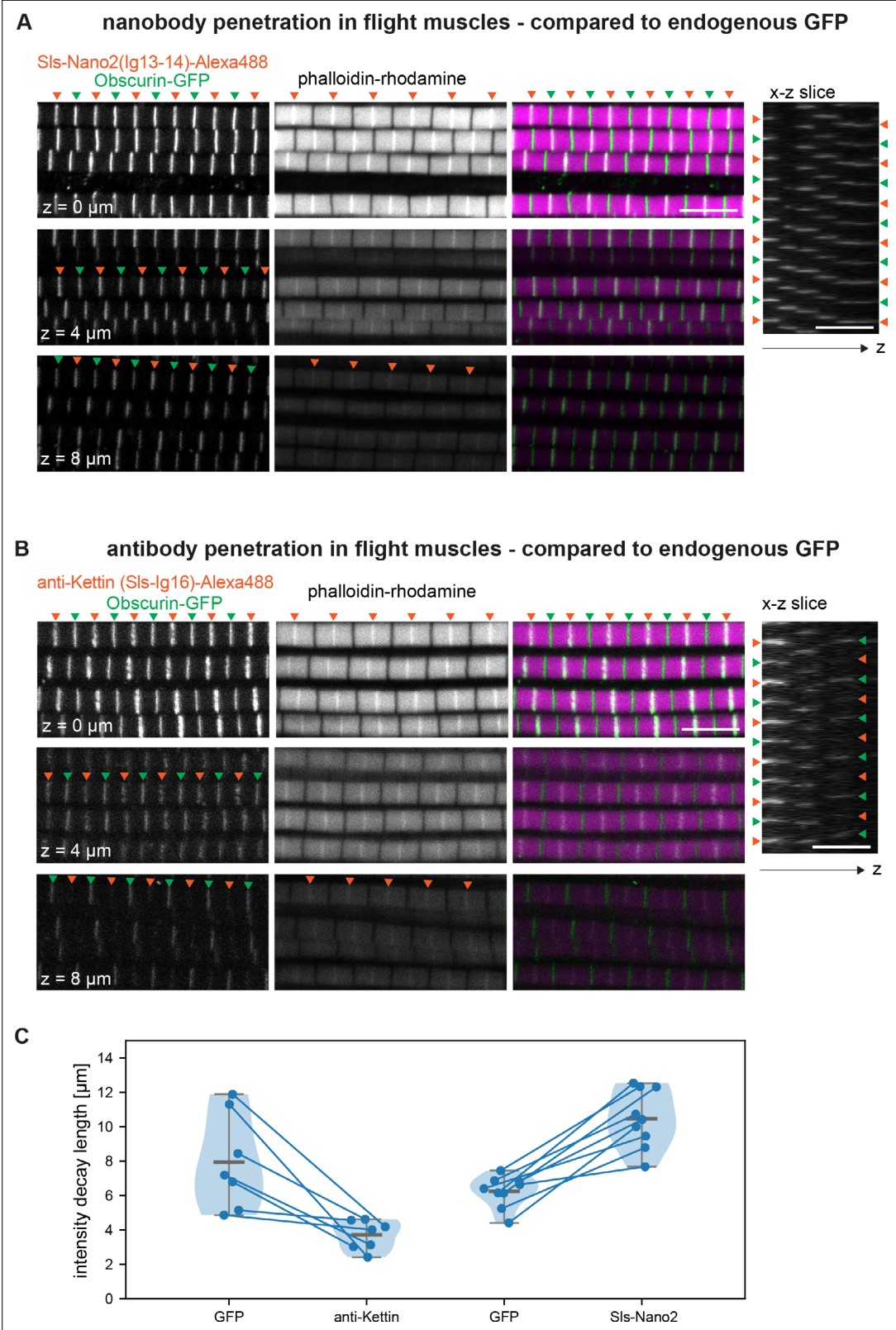

**Figure 6.** Nanobody penetration. (**A and B**) Scanning confocal images of adult hemithorax expressing Obscurin-GFP (green arrowheads) in flight muscles stained with phalloidin to label actin (magenta) and either Sls-Nano2-Alexa488 (**A**) or anti-Kettin antibody (binding Sls-Ig16) (red arrowheads), followed by secondary antibody coupled with Alexa488 (**B**). Three different z-planes and x-z slices are shown. Note that nanobody (red arrowheads in A) and GFP signals (green arrowheads) are visible in the entire z-stack, whereas the antibody signal decays quickly in the z-direction (red arrowheads

*Figure 6 continued on next page*

*Figure 6 continued*

in B). Scale bars 5 µm. (**C**) Fluorescence detection decay length versus imaging depth for GFP, anti-Kettin and Sls-Nano2 (anti-Kettin vs. Sls-Nano2 comparison: p-value = 0.0001748, Mann-Whitney test).

The online version of this article includes the following source data and figure supplement(s) for figure 6:

**Source data 1.** Source data for *Figure 6*.

**Figure supplement 1.** Quantification of intensity decay.

**Figure supplement 2.** Nanobody penetration in flight muscles.

**Figure supplement 3.** Nanobodies penetrate embryos easier than antibodies.

technical artefact as double staining with Proj-Nano30 and Proj-Nano28 or with Proj-Nano35 and Proj-Nano46 showed an almost perfect overlap (*Figure 8—figure supplement 1E, F*). Finally, we confirmed the 'shifted-blocks' pattern by imaging the Proj-Nano30(Ig5-8) and Proj-Nano37(kinase) patterns with an airy-scan detector that slightly increases the spatial resolution (*Figure 8C*).

We hypothesised that the small central gap visible in the Projectin kinase domain nanobody patterns is caused by a Projectin-free M-band of the larval sarcomere. Co-labelling the M-band with our anti-Obscurin nanobody Obs-Nano58 confirmed that the gap present in the Proj-kinase nanobody pattern is consistent with its absence from the M-band (*Figure 8D*). Taken together, our results demonstrate that Projectin decorates the myosin filaments in a defined polar orientation, likely from the tip of the myosin filaments until the beginning of the H-zone that is devoid of myosin heads (*Figure 8E*).

## Live imaging of Sls using nanobodies *in vivo*

Nanobodies have the particular advantage that they are single-chain proteins that can be expressed in the cytoplasm of eukaryotic cells. To our knowledge, only nanobodies against GFP, mCherry, or short epitope tags had thus far been expressed in *Drosophila* tissues (*Caussinus et al., 2011*; *Harmansa and Affolter, 2018*; *Harmansa et al., 2015*; *Harmansa et al., 2017*; *Xu et al., 2022*). Hence, we wanted to test if our nanobodies are useful tools to track a native sarcomeric protein in the mature muscle, similar to a direct GFP fusion to the sarcomeric protein. For proof of principle experiments, we chose Sls-Nano2 for two reasons: first, Sls is likely to be stably incorporated into mature sarcomeres, and its large size should prevent fast diffusion. Thus, Sls is a suitable protein to test if a nanobody would be stably bound to a target protein in muscle. Second, we verified the high affinity of Sls-Nano2 to the Sls-Ig13/14 target *in vitro* (*Figure 3B*).

We first tested if the expression of Sls-Nano2-NeonGreen has any deleterious effects on the developing muscles. We expressed the nanobodies with the muscle-specific *Mef2*-GAL4 driver and fixed the larvae to assay the morphology of larval muscles with anti-Sls nanobodies Sls-Nano2 and Sls-Nano42. We found that the sarcomere morphology and the length of the Sls protein are normal (*Figure 9—figure supplement 1A, B*). To assess muscle function, we placed L3 larvae on an agar plate and recorded their locomotion using standard software (*Risse et al., 2017*). We found that larvae expressing Sls-Nano2-NeonGreen during muscle development move with a comparable speed and persistence as controls (*Figure 9—figure supplement 1C, D*, *Figure 9—videos 1 and 2*). Hence, Sls-Nano2-NeonGreen expressing larvae are a good tool to investigate Sls dynamics *in vivo*.

To test if Sls-Nano2-NeonGreen can indeed visualise Sls *in vivo*, we assayed muscles of intact living L3 larvae expressing Sls-Nano2-NeonGreen under *Mef2*-GAL4 control. We found the expected striated pattern of Sls-Nano2-NeonGreen labelling a single thin stripe, resembling the Sls-Nano2 staining in fixed larval muscles (*Figure 9A*). We conclude that Sls-Nano2-NeonGreen binds to Sls-Ig13/14 *in vivo*.

Encouraged by these results, we also generated NeonGreen-fusions for three other nanobodies and found the expected patterns for Sls-Nano42-NeonGreen, two defined bands at about 2 µm distance from the Z-disc and for Proj-Nano30-NeonGreen or Proj-Nano37-NeonGreen, two thick blocks right and left to the M-band, when expressed in larval muscles (*Figure 9A*). Thus, the *in vivo* expressed nanobodies bind their target epitopes in living muscles as they do in fixed muscles.

To quantify the diffusion and local turnover of Sls-Nano2-NeonGreen, we adapted a protocol that allowed us to image intact living larvae under the spinning disc microscope for at least 30 min (see Methods) (*Kakanj et al., 2020*). This enabled us to measure fluorescence recovery after photobleaching (FRAP) of Sls-Nano2-NeonGreen in living larval muscles. We bleached one area in L3 larval

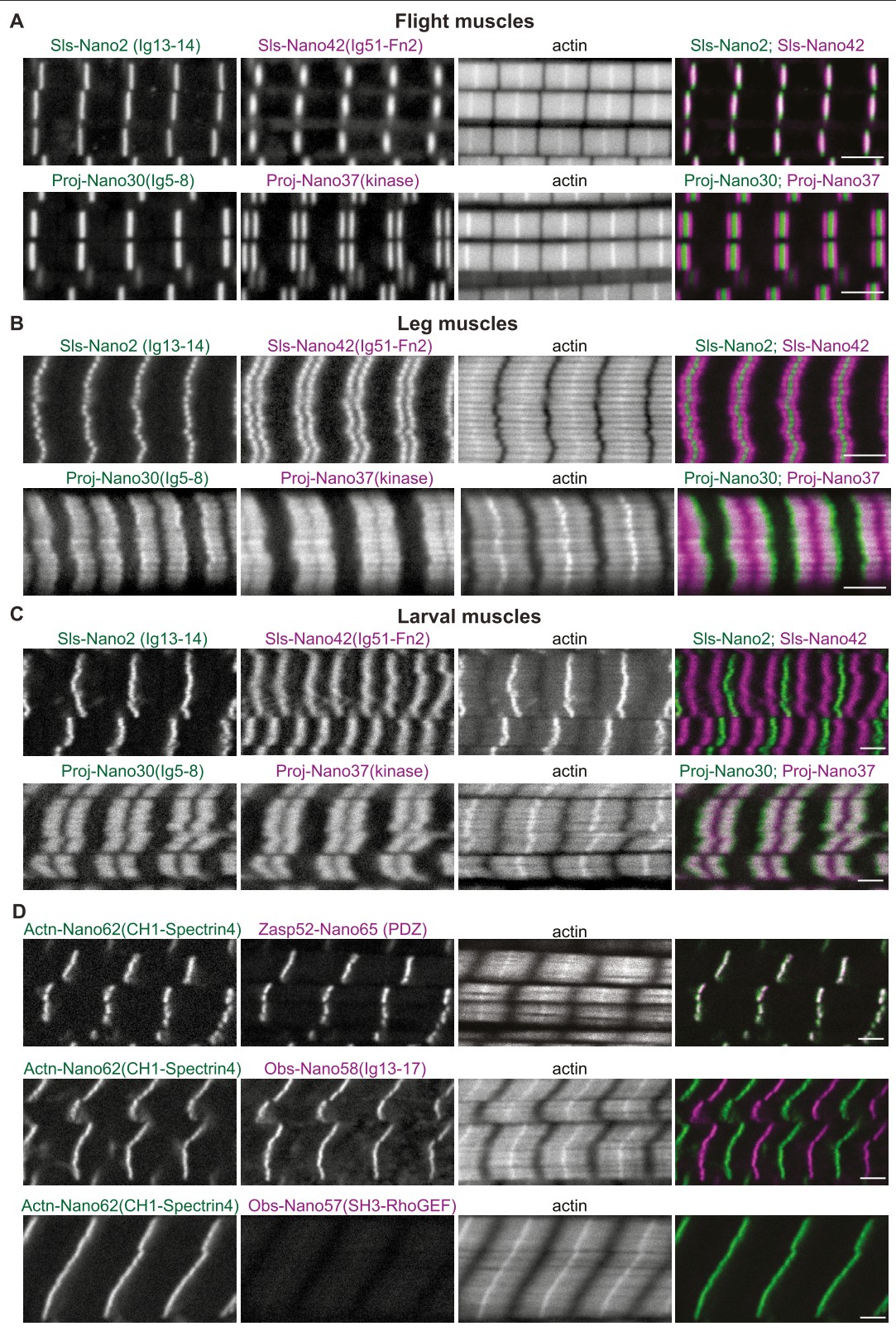

**Figure 7.** Sallimus, Projectin, α-Actinin, Obscurin, and Zasp52 localisation in different mature sarcomere types. (**A–C**) Scanning confocal images of mature sarcomeres from wild-type flight muscles (**A**), leg muscles (**B**) or L3 larval muscles (**C**) stained by phalloidin (actin) together with N- and C-terminal anti-Sls nanobodies (Sls-Nano2 in green and Sls-Nano42 in magenta, top) or N- and C-terminal anti-Projectin nanobodies (Proj-Nano30 in green and Proj-Nano37 in magenta, bottom). Scale bars are 3 μm. Note the long distance between the Sls-Nano42 bands in larval muscles and the distinct

*Figure 7 continued on next page*

Figure 7 continued

locations of Proj-Nano30 and Proj-Nano37 in leg and larval muscles. (**D**) L3 larval muscles stained by phalloidin (actin) together with anti-Actn and anti-Zasp52 nanobodies (Actn-Nano62 in green and Zasp52-Nano65 in magenta, top), or with anti-Actn and anti-Obscurin nanobodies (Actn-Nano62 in green and Obs-Nano58 in magenta, middle and Actn-Nano62 in green and Obs-Nano57 in magenta, bottom). Note the absence of the Obscurin SH3-RhoGEF domain signal from the larval muscle. Scale bars are 3 μm.

The online version of this article includes the following figure supplement(s) for figure 7:

**Figure supplement 1.** Sallimus and Projectin localisation in mature flight muscle.

muscles and measured fluorescence recovery over 29 min (*Figure 9B and C*, *Figure 9—video 3*). We found very little recovery during the observation period. This demonstrates that the Sls-Nano2 is indeed stably bound to Sls-Ig13/14 target and that Sls protein does not exchange significantly over a 30-min period in mature larval muscles. Together, these data verified that nanobodies against *Drosophila* proteins can indeed bind their target *in vivo* and thus can be used to investigate the dynamics of a chosen target domain. Hence, the here generated nanobodies will be invaluable tools to quantify the dynamics of Sls and Projectin during muscle development and homeostasis.

## Degradation of Sls protein in muscles *in vivo*

A nanobody against GFP was already previously fused to a degradation signal to degrade GFP-fusion proteins in *Drosophila* cells *in vivo* (*Caussinus et al., 2011*; *Harmansa and Affolter, 2018*; *Nagarkar-Jaiswal et al., 2015*). This is a widely useful strategy; however, it needs functional GFP-fusion proteins and complex genetics to combine nanobody and GAL4-driver with the homozygous GFP-fusion. To test if our nanobodies could be engineered to induce degradation of Sallimus, we fused the same F-box as used for the GFP nanobodies (NSlmb) (*Caussinus et al., 2011*) to Sls-Nano2 and Sls-Nano42 and made transgenic flies with them under UAS control that we called UAS-Sls-Nano2-deGrad and UAS-Sls-Nano42-deGrad, respectively. When expressing Sls-Nano2-deGrad under *Mef2*-GAL4 control in larval muscles from embryonic stages onwards, we found in stainings that the fluorescent signal of labelled Sls-Nano2 (Ig13/14) was reduced by ~80% as compared to control larvae. However, staining with the anti-Kettin antibody (binding to the neighbouring Sls-Ig16) was less reduced, whereas C-terminal Sls-Nano39 showed a normal intensity (*Figure 10A and B*). We interpret this pattern as Sls-Nano2-deGrad partially masking its epitope and as an incomplete, segment-wise degradation of Sallimus. The incompleteness of degradation can be explained by a stable sarcomeric assembly limiting the access of the proteasome and by the *Mef2* promoter (driving expression of the deGrad-Nanobody) being weaker than the Sallimus promoter in mature larval muscles.

We found that expression of Sls-Nano2-deGrad in muscles caused a drastic phenotype later in development, namely lethality at the pupal stage; no adults were eclosing (*Figure 10C*). Similarly, the majority of the Sls-Nano42-deGrad expressing pupae died, since much fewer than the expected 50% of adults eclosed from the cross of the heterozygous line to the *Mef2*-GAL4 driver (*Figure 10C*).

The few surviving Sls-Nano42-deGrad adults showed again a reduction in Sls-staining intensities with a nearly complete signal loss of Sls-Nano42 and a ~50% reduction with the neighbouring Sls-Nano39. The distant Sls-Nano2 epitope showed no reduction (*Figure 10D and E*). Again, this suggests a segment-wise Sallimus degradation in the flight muscle. In addition, we observed a variable myofibril thickness in the few surviving Sls-Nano42-deGrad adults, which is never found in wild type (*Figure 10D*), pointing to a specific defect in myofibril maturation that might be controlled by Sls availability.

## Discussion

### Nanobodies as tools for developmental biology

Thus far, the application of nanobodies in *Drosophila* had been limited to nanobodies against fluorescent proteins or recently against short epitope tags (*Caussinus et al., 2011*; *Harmansa and Affolter, 2018*; *Harmansa et al., 2015*; *Harmansa et al., 2017*; *Xu et al., 2022*). These former studies have shown that nanobodies against GFP can be used to trap secreted Dpp in the *Drosophila* wing disc and hence demonstrated the strong binding of nanobodies to their target also *in vivo* (*Harmansa et al., 2015*; *Harmansa et al., 2017*). Here we demonstrated that the high affinity of nanobodies to their

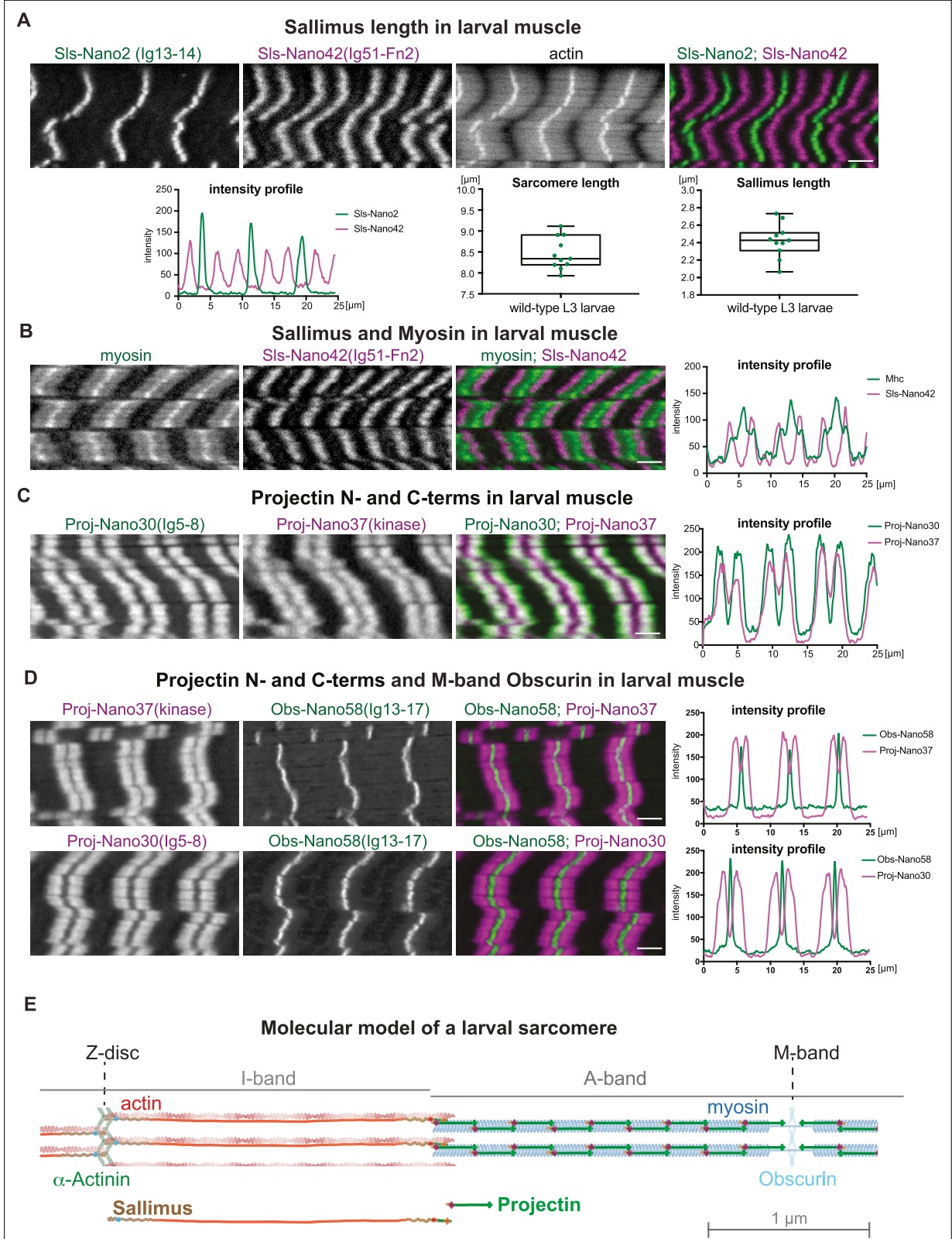

**Figure 8.** Sallimus and Projectin localisation patterns in mature larval sarcomeres. (**A**) Scanning confocal images of L3 larval muscles stained for actin, and N-(Sls-Nano2, green) and C-terminal (Sls-Nano42, magenta) anti-Sls nanobodies. Scale bar 3 μm. Plot displays longitudinal intensity profiles of Sls-Nano2 and Sls-Nano42. Quantification of sarcomere length (distance between Sls-Nano2 bands) and Sls length (distance between Sls-Nano2 and Sls-Nano42). Each point represents an animal, *n* = 10. (**B**) L3 larval muscle stained for myosin (green) and C-terminal (Sls-Nano42, magenta) anti-Sls

*Figure 8 continued on next page*

*Figure 8 continued*

nanobody. Scale bar 3 µm. Plot displays intensity profiles of myosin and Sls-Nano42. Note that peaks of Nano42 map to the start of the myosin signal. (**C**) L3 larval muscle stained for anti-Projectin with N-(Proj-Nano30, green) and C-terminal (Proj-Nano37, magenta) nanobodies and imaged with an airy-scan detector. Scale bar 3 µm. Plot displays intensity profiles of Proj-Nano30 and Proj-Nano37. Note that the Proj-Nano37 signal is closer to the M-band compared to Proj-Nano30. (**D**) L3 larval muscle stained for Obscurin (Obs-Nano58, green) and Projectin either with N-(Proj-Nano30, magenta) or C-terminal (Proj-Nano37, magenta) nanobodies and imaged with an airy scan detector. Scale bar 3 µm. Plots display intensity profiles. Note that Obscurin perfectly fills the Proj-Nano37 gap at the M-band. (**E**) Molecular model of a larval sarcomere. Note the extended Sallimus across the I-band and the staggered Projectin on the myosin filaments in the A-band leaving the M-band free.

The online version of this article includes the following source data and figure supplement(s) for figure 8:

**Source data 1.** Source data of *Figure 8*.

**Figure supplement 1.** Sallimus and Projectin localisation in larval muscle.

targets *in vivo* is not limited to the commercially available GFP nanobody that the fly community has extensively used in the past (*Caussinus et al., 2011*; *Harmansa and Affolter, 2018*). This is significant as many GFP fusion proteins do not retain full functionality, as reported not only for Dpp-GFP but also for sarcomeric proteins such as Mhc-GFP, Sls-GFP, or troponin-GFP fusion attempts (*Matsuda et al., 2021*; *Orfanos et al., 2015*; *Sarov et al., 2016*).

High affinity to the target epitopes is likely the case for most members of the here presented nanobody toolbox, as exemplified in detail for Sls-Nano2. Our FRAP data of Sls-Nano2-NeonGreen suggest that Sallimus is not mobile in a 30-min interval in mature larval sarcomeres. It will be interesting to extend these studies to longer time frames as mammalian titin was suggested to be surprisingly dynamic at least in *in vitro* cultured cardiomyocytes (*Rudolph et al., 2019*).

Nanobodies can also be engineered to induce the degradation or inactivation of the target protein *in vivo* (*Caussinus et al., 2011*; *Nagarkar-Jaiswal et al., 2015*). Our proof of principle experiments presented here suggests that this is likely also the case for the here developed Sls nanobodies when fused to a degradation signal. However, how sarcomeric proteins are turned over is still unclear. Given that sarcomeres are very dense structures that likely exclude most soluble proteins (*O'Donnell et al., 1989*), it is hard to imagine how 26S proteasomes can gain access. Another challenge is how to degrade a protein as large as Sallimus. A significant force would be required to pull it out of its sarcomeric anchorage since Sls is bound to the Z-disc at one end and to the thick filament at the other. It is thus perhaps not surprising that targeting a single degron to Sallimus does not suffice for a complete degradation. The induced lethality during pupal stages, however, suggests some dramatic consequences during adult muscle development that need further analysis.

Nanobodies were also used as conditional blockers of their target domains, such as blocking the kinase domain of estimated glomerular filtration rate in cell culture (*Tabtimmai et al., 2019*), even without a degradation signal. Hence, the here generated nanobody toolbox is a first step towards a modulation of Sls or Projectin domain activity *in vivo*.

The small size of nanobodies not only allows superior penetration into tissues as shown here for late-stage *Drosophila* embryos or thick flight muscle tissue but also places possible labels very close to their target epitopes. This is relevant for super-resolution microscopy that can resolve the target location with a precision better than 5 nm resolution (*Ostersehlt et al., 2022*; *Schnitzbauer et al., 2017*) or for cryo-electron-tomography, with which the native structure of titin in the sarcomere might be resolvable in the future (*Wang et al., 2022*; *Wang et al., 2021*). High labelling density and proximity of the label to the target are key to identify the nature of unknown protein densities in tomograms. Hence, our toolbox should not only provide a resource to mechanistically study the function of the *Drosophila* sarcomeric proteins in more detail in the future but may also inspire the *Drosophila* community to invest more into the generation of nanobodies, instead of generating antibodies by default.

## A *Drosophila* titin and sarcomere nanobody toolbox

We introduced here the generation and characterisation of 22 different nanobodies that were raised against 11 different target domains, three are present in Sls and four in Projectin, two in Obscurin and one each in α-Actinin and Zasp52. We characterised their specificity in embryonic and larval muscles and verified that nanobodies are indeed well suited to diffuse into dense muscle tissues. They even

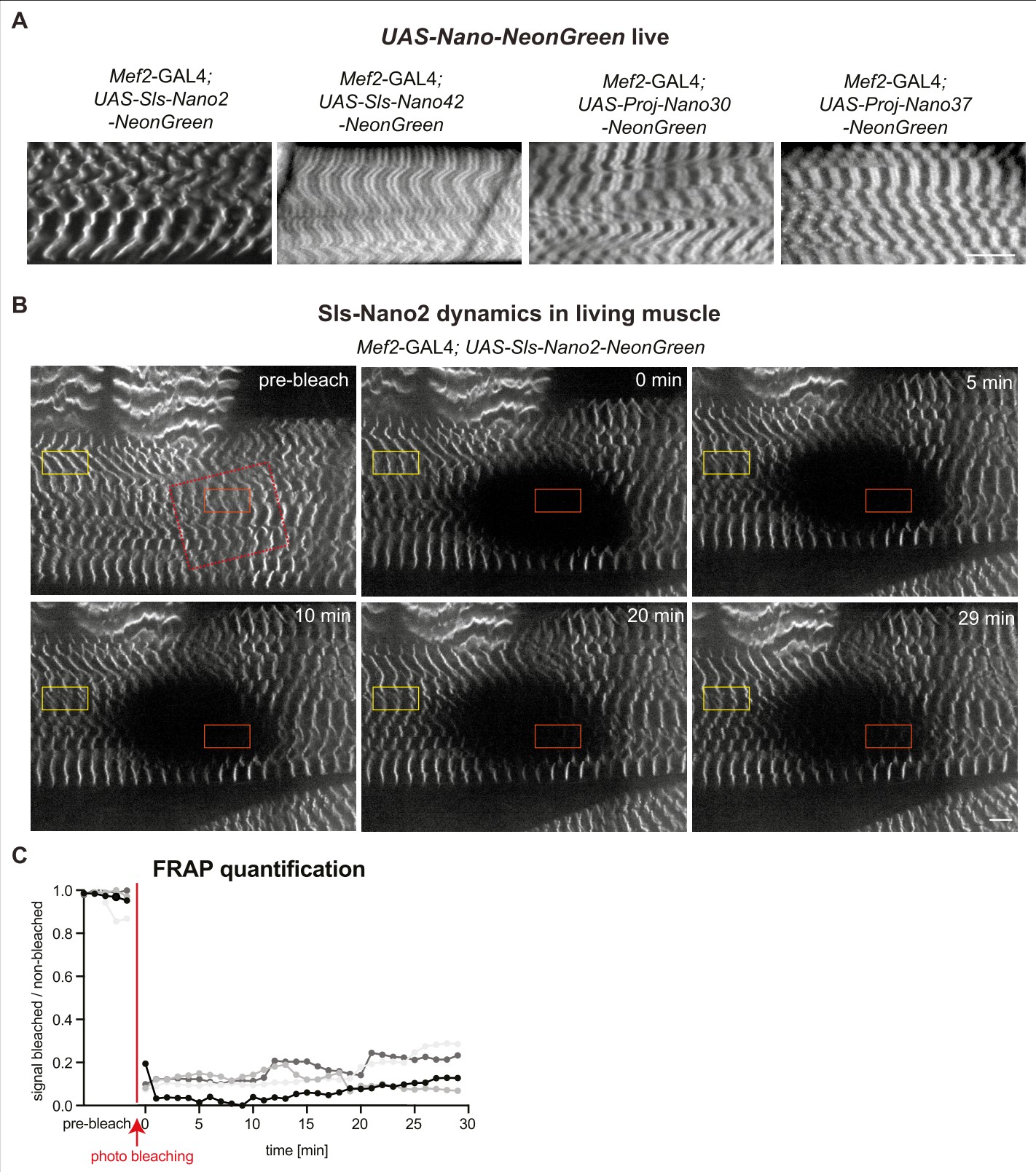

**Figure 9.** Live imaging of Sls using nanobodies *in vivo*. (**A**) Live imaging of *UAS-Nano-NeonGreen* expressing larvae (*Mef2*-GAL4) with spinning disc microscopy. NeonGreen was fused to Sls-Nano2, Sls-Nano42, Proj-Nano30 or Proj-Nano37 nanobodies. Note the thin single stripes of Sls-Nano2-NeonGreen and the 2 stripes of Sls-Nano42-NeonGreen at the expected distance. Proj-Nano30-NeonGreen shows thick blocks that are further away from the M-band than the thick Proj-Nano37-NeonGreen blocks. Scale bar 20 µm. (**B**) Living L3 larval muscles expressing Sls-Nano2-NeonGreen

*Figure 9 continued on next page*

*Figure 9 continued*

expressed with *Mef2*-GAL4 imaged with spinning disc microscopy. Note the striated pattern marking the Z-discs. A region marked by the red rectangle was bleached (the larva was slightly moving while being bleached) and fluorescence recovery was imaged. A single z-plane of a stack is shown. Scale bar 10 µm. (**C**) Quantification of fluorescence recovery in the orange box, which was normalised by the fluorescence in the yellow box outside the bleached area. The different grey values indicate four different larvae from four different experiments. Note either absence or less than 20% recovery in the bleached area over 30 min.

The online version of this article includes the following video, source data, and figure supplement(s) for figure 9:

**Source data 1.** Source data for *Figure 9*.

**Figure supplement 1.** Nanobody-NeonGreen fusions *in vivo*.

**Figure 9—video 1.** Larval crawling and tracking in wild type.

https://elifesciences.org/articles/79343/figures#fig9video1

**Figure 9—video 2.** Larval crawling and tracking in Sls-Nano2-NeonGreen larvae.

https://elifesciences.org/articles/79343/figures#fig9video2

**Figure 9—video 3.** Live imaging of Sls with a nanobody using fluorescence recovery after photobleaching.

https://elifesciences.org/articles/79343/figures#fig9video3

label muscles of late stage embryos, which are impermeable to antibodies because of their chitin skeleton (*Moussian, 2010*).

Staining larval, leg and flight muscles with our nanobodies confirmed the existence of different Sls, Projectin and Obscurin isoforms in the different muscle types. The stiff flight muscles do contain a short version of Sls, which does not allow to resolve the N- and C-terminal ends of Sls using confocal microscopy. This was only possible by using super-resolution microscopy with our here developed nanobodies (*Schueder et al., 2023*). Larval muscles express a novel shorter Obscurin isoform missing the N-terminal SH3 and RhoGEF domains.

Our data suggest that most of the Sls isoforms present in flight muscles do contain the C-terminal Sls-Ig51-Fn2 domains. This is consistent with developmental transcriptomics results that included splice isoform annotations (*Spletter et al., 2015*; *Spletter et al., 2018*), and the very low expression of a Sls isoform that uses an early alternative stop codon, which is rather expressed in leg muscles (*Sarov et al., 2016*). This is significant as the initially proposed short Sls isoform named Kettin is not supposed to contain the C-terminal Sls-Ig51-Fn2 domains and hence would not bridge across the thin I-band of flight muscles to the myosin filament (*Burkart et al., 2007*; *Lakey et al., 1993*; *Szikora et al., 2020*). Our new nanobodies now clarify that most Sls isoforms have at least the potential to bridge to the myosin filament in flight muscles (*Schueder et al., 2023*).

Similarly, our Projectin nanobodies verified the defined orientation of elongated Projectin in flight muscle sarcomeres with its N-terminus facing the Z-disc and its C-terminal kinase domain oriented towards the centre of myosin filament. In the accompanying manuscript, these tools enabled the determination of the precise position of the Projectin ends in the flight muscles using super-resolution microscopy (*Schueder et al., 2023*).

## A long stretched Sls isoform in larval muscle

Larval muscles are considered soft compared to stiff flight muscles. This is consistent with their large dynamic length range: larval sarcomeres have a relaxed length of about 8.5 µm and can contract up to about 4.5 µm. In contrast, flight muscle sarcomeres only contract 3.5% of their length during flight, about 120 nm (measured in *Drosophila virilis Chan and Dickinson, 1996*). Consistent with this, the I-band of relaxed larval muscles is long, about 2 µm. Hence, our finding that Sls has a length of more than 2 µm in relaxed larval muscles is only logical, considering that Sls needs to elastically bridge from the Z-disc to the myosin filament. However, this finding still comes as a significant surprise, since the mammalian titin is considered to be the 'longest' protein in the animal kingdom; however, it is 'only' 1.5 µm long in 3 µm long relaxed human sarcomeres (*Linke, 2018*; *Llewellyn et al., 2008*; *Regev et al., 2011*). Mammalian titin is certainly the largest protein by molecular weight (up to 3800 kDa) (*Brynnel et al., 2018*), whereas the longest predicted *Drosophila* Sls isoform has a mass of 'only' 2100 kDa. The long extension of Sls found here makes it likely that Sls is under strong mechanical tension in larval muscles, and hence its long PEVK spring domains are likely unfolded to allow bridging of the long I-band in the relaxed state of the larval muscle (model in *Figure 8E*). Such, *Drosophila* Sls

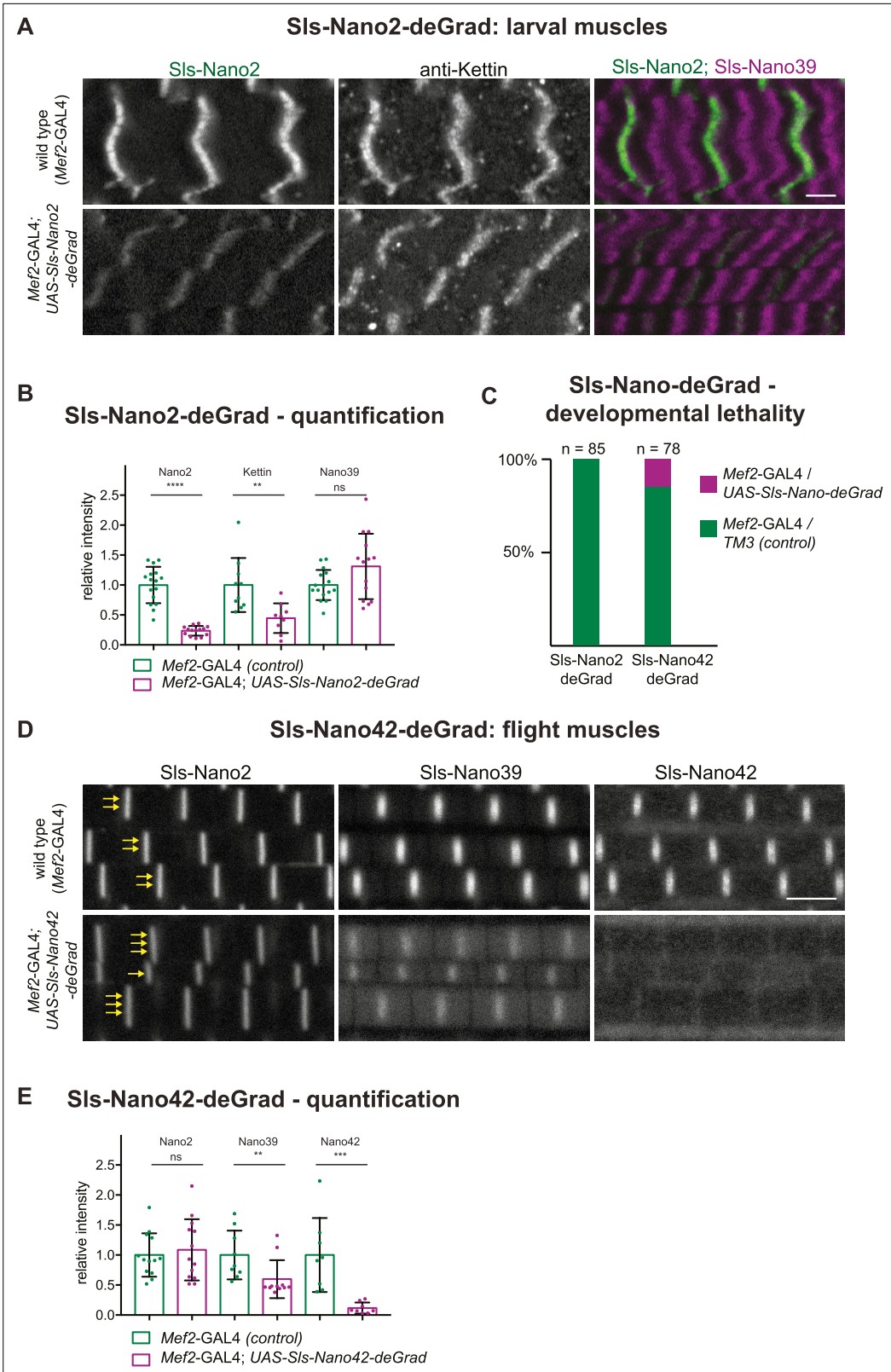

**Figure 10.** Sls-Nano-deGrad *in vivo*. (**A**) Control (top) or Sls-Nano2-deGrad (bottom) expressing larval muscles stained with Sls-Nano2 (green), anti-Kettin (white), and Sls-Nano39 (magenta) and imaged with scanning confocal microscopy. Scale bar is 3 µm. Note the strong reduction of Sls-Nano2 signal, whereas anti-Kettin is weakly reduced. (**B**) Quantifications of staining intensities shown in (**A**). Each dot represents an animal (p-value

*Figure 10 continued on next page*

*Figure 10 continued*

Nano2:<0.0001; Kettin = 0.002; Nano39 = 0.1279; Mann–Whitney test). (**C**) Lethality assay upon crossing *UAS-Sls-Nano-deGrad/TM3* to *Mef2*-GAL4. Note the complete lethality induced by the expression of Sls-Nano2-deGrad and the partial lethality induced by Sls-Nano42-deGrad (expected ratio to *Mef2*-GAL4/*TM3* control: 50%). (**D**) Control (top) or Sls-Nano42-deGrad (bottom) expressing adult flight muscles stained with Sls-Nano2, Sls-Nano39, and Sls-Nano42. Note the strong reduction of Sls-Nano42 signal and the weaker reduction of Sls-Nano39. Note the irregular thickness of the myofibrils in Sls-Nano42-deGrad flight muscles compared to control indicated by varying number of yellow arrows. Scale bar is 3 μm. (**E**) Quantifications of staining intensities shown in (**D**). Each dot represents an animal (p-value Nano2 = 0.8288; Nano39 = 0.004; Nano42 = 0.0002; Mann–Whitney test).

The online version of this article includes the following source data for figure 10:

**Source data 1.** Source data for *Figure 10*.

---

may store a significant amount of energy for the next round of muscle contraction, purely by unfolding its PEVK domains and not necessarily needing to unfold any of its Ig domains, the latter has recently been suggested for mammalian titin (*Rivas-Pardo et al., 2016*; *Rivas-Pardo et al., 2020*). Hence, our data identify that insect Sls might indeed be one of the 'longest' proteins naturally occurring in animals, a truly deserving member of the titin protein family. Similarly, long titin family members with extensive elastic domains were also found in the claw muscles of crayfish (as *Drosophila* an arthropod) that contain extensively long sarcomeres (*Fukuzawa et al., 2001*). Importantly, also *C. elegans* contains a Sallimus homolog called TTN-1. TTN-1 is a 2 MDa protein that spans across the I-band and was thus also suggested to mechanically link the Z-disc to the myosin filaments in *C. elegans* body muscles (*Forbes et al., 2010*).

## A defined orientation of Projectin on myosin filaments

In contrast to Sls, Projectin does not locate in a sharp band in larval or leg muscles, but rather as a broad block, which had been previously reported (*Lakey et al., 1990*; *Saide et al., 1989*; *Vigoreaux et al., 1991*). Our data revealed here that the N- and C-terminal ends of Projectin display slightly shifted localisations, with the C-terminus located closer to the M-band compared to the N-terminus. This strongly suggests that each Projectin protein has a defined orientation on the myosin filament (model in *Figure 8E*). Currently, it remains unclear if neighbouring Projectin molecules overlap or if they are arrayed in a linear way to decorate the thick filament, similar to how the mammalian titin protein is supposed to decorate it (*Tonino et al., 2017*). If they do not overlap much, about eight or nine ~250-nm long Projectin molecules (*Schueder et al., 2023*) would be needed to bridge half the myosin filament. This is consistent with the theoretical length of a chain of about 70 folded Ig and Fn domains, each about 4 nm (*Mayans et al., 2001*). Super-resolution imaging of larval muscles using the here generated nanobodies will be needed to answer this interesting question.

The fact that Projectin decorates the entire thick filament of likely all *Drosophila* muscles, except indirect flight muscles, has the interesting consequence that the Projectin kinase is also located along the entire thick filament. The same is true for the *C. elegans* Projectin homolog called Twitchin (*Forbes et al., 2010*), so *Drosophila* Projectin is not the exception. Titin kinases, including the Projectin kinase, are possibly modulated by mechanical stretch: an inhibitory C-terminal tail needs to be pulled out of the kinase domain to allow kinase activity (*Gautel, 2011*; *Gräter et al., 2005*; *Kobe et al., 1996*; *Lange et al., 2005*). Thus, the larval muscle localisation of Projectin would allow it to respond to stretch with kinase activation along the entire thick filament, and not only at the M-band as is the case for mammalian muscle. Hence, it will be interesting to test if the Projectin kinase activity is required for sarcomere formation or function. Thus far, this has been tested in *C. elegans*: a kinase-dead variant of Twitchin results in normal sarcomere morphology but abnormally strong muscle contractions that lead to an evolutionary disadvantage (*Matsunaga et al., 2017*). The *Drosophila* larval muscles would be a good model to further investigate the role of this evolutionarily conserved kinase domain, and our here generated nanobodies, four of which target the Projectin kinase domain might be a valuable tool for such future studies.

# Methods

### Key resources table

| Reagent type (species) or resource | Designation | Source or reference | Identifiers | Additional information |
|---|---|---|---|---|
| Strain, strain background (*Drosophila melanogaster*) | Luminy | **Leonte et al., 2021** | | |
| Strain, strain background (*Drosophila melanogaster*) | *Mef2*-GAL4 | **Schnorrer et al., 2010** | | |
| Strain, strain background (*Drosophila melanogaster*) | sls-IR (TF47301) | **Dietzl et al., 2007** | | |
| Strain, strain background (*Drosophila melanogaster*) | UAS-bt-IR (TF46252) | **Dietzl et al., 2007** | | |
| Strain, strain background (*Drosophila melanogaster*) | UAS-Actn-IR (TF7760) | **Dietzl et al., 2007** | | |
| Strain, strain background (*Drosophila melanogaster*) | UAS-Zasp52-IR (JF01133) | **Ni et al., 2011** | | |
| Strain, strain background (*Drosophila melanogaster*) | Unc-89[EY15484] | **Katzemich et al., 2012** | | |
| Gene (*Drosophila melanogaster*) | sls | http://flybase.org/reports/FBgn0086906 | FBgn0086906 | |
| Gene (*Drosophila melanogaster*) | bt (Projectin) | http://flybase.org/reports/FBgn0005666 | FBgn0005666 | |
| Gene (*Drosophila melanogaster*) | Unc-89 (Obscurin) | http://flybase.org/reports/FBgn0053519 | FBgn0053519 | |
| Gene (*Drosophila melanogaster*) | Actn | http://flybase.org/reports/FBgn0000667 | FBgn00006679 | |
| Gene (*Drosophila melanogaster*) | Zasp52 | http://flybase.org/reports/FBgn0265991 | FBgn0265991 | |
| Antibody | anti-Mhc (Mouse monoclonal) | DHSB | 3e8-3D3 | IF(1:100) |
| Antibody | anti-Sls (Kettin) (Rat monoclonal) | Babraham Institute | MAC155 | IF(1:500) |
| Antibody | anti-Projectin (Rat monoclonal) | Babraham Institute | MAC150 | IF(1:100) |
| Other | Sls-Ig13/14 (Nano2) | This study | Coupled to A488; A647; STAR RED | Nanobody – used at about 50 nM; see Materials availability statement |
| Other | Sls-Ig49/50 (Nano39) | This study | Coupled to A488; A647; | Nanobody – used at about 50 nM; see Materials availability statement |
| Other | Sls-Ig51-Fn2 (Nano42) | This study | Coupled to A488; A647; | Nanobody – used at about 50 nM; see Materials availability statement |
| Other | Sls-Ig51-Fn2 (Nano48) | This study | Coupled to A488; A647; | Nanobody – used at about 50 nM; see Materials availability statement |
| Other | Proj-Fn1/2 (Nano28) | This study | Coupled to A488; A647; | Nanobody – used at about 50 nM; see Materials availability statement |
| Other | Proj-Fn1/2 (Nano29) | This study | Coupled to A488; A647; | Nanobody – used at about 50 nM; see Materials availability statement |
| Other | Proj-Ig5-8 (Nano30) | This study | Coupled to A488; A647; | Nanobody – used at about 50 nM; see Materials availability statement |
| Other | Proj-Ig27-Fn35 (Nano33) | This study | Coupled to A488; A647; | Nanobody – used at about 50 nM; see Materials availability statement |
| Other | Proj-kinase (Nano34) | This study | Coupled to A488; A647; | Nanobody – used at about 50 nM; see Materials availability statement |
| Other | Proj-kinase (Nano35) | This study | Coupled to A488; A647; | Nanobody – used at about 50 nM; see Materials availability statement |

| Reagent type (species) or resource | Designation | Source or reference | Identifiers | Additional information |
|---|---|---|---|---|
| Other | Proj-kinase (Nano37) | This study | Coupled to A488; A647; | Nanobody – used at about 50 nM; see Materials availability statement |
| Other | Proj-kinase (Nano46) | This study | Coupled to A488; A647; | Nanobody – used at about 50 nM; see Materials availability statement |
| Other | Obscurin-SH3-RhoGEF (Nano55) | This study | Coupled to A488; A647; | Nanobody – used at about 50 nM; see Materials availability statement |
| Other | Obscurin-SH3-RhoGEF (Nano56) | This study | Coupled to A488; A647; | Nanobody – used at about 50 nM; see Materials availability statement |
| Other | Obscurin-SH3-RhoGEF (Nano57) | This study | Coupled to A488; A647; | Nanobody – used at about 50 nM; see Materials availability statement |
| Other | Obscurin-Ig13-17 (Nano58) | This study | Coupled to A488; A647; | Nanobody – used at about 50 nM; see Materials availability statement |
| Other | Obscurin-Ig13-17 (Nano59) | This study | Coupled to A488; A647; | Nanobody – used at about 50 nM; see Materials availability statement |
| Other | α-Actinin-CH1-Spec4 (Nano62) | This study | Coupled to A488; A647; | Nanobody – used at about 50 nM; see Materials availability statement |
| Other | α-Actinin-CH1-Spec4 (Nano63) | This study | Coupled to A488; A647; | Nanobody – used at about 50 nM; see Materials availability statement |
| Other | α-Actinin-CH1-Spec4 (Nano64) | This study | Coupled to A488; A647; | Nanobody – used at about 50 nM; see Materials availability statement |
| Other | Zasp52-PDZ (Nano65) | This study | Coupled to A488; A647; | Nanobody – used at about 50 nM; see Materials availability statement |
| Other | Zasp52-PDZ (Nano66) | This study | Coupled to A488; A647; | Nanobody – used at about 50 nM; see Materials availability statement |
| Chemical compound, drug | Rhodamine-phalloidin | Invitrogen, Cat. R415 | | 1 in 500 |

## Recombinant immunogens and nanobody generation

We screened existing transcriptomics data (*Spletter et al., 2015*; *Spletter et al., 2018*) and Flybase (http://flybase.org/reports/FBgn0086906; http://flybase.org/reports/FBgn0005666) to identify candidate domains of Sls and Projectin that should be expressed in all or most muscle types. Next, we used Swissmodel (*Waterhouse et al., 2018*) to predict domain borders for stably folding fragments. These fragments were then codon-optimised for expression in *E. coli* and cloned into a His14-bdSUMO fusion vector (*Frey and Görlich, 2014a*). Expression was in *E. coli* NEB Express Iᑫ at 21 °C, in 2YT + 50 µg/ml kanamycin with 4 hr of induction with 100 µM isopropyl β- d-1-thiogalactopyranoside (IPTG). Bacteria were pelleted by centrifugation, resuspended in 50 mM Tris/HCl pH 7.5, 20 mM imidazole/HCl pH 7.5, 300 mM NaCl, and lysed by a freeze-thaw cycle followed by sonication. The lysate was cleared by ultracentrifugation in a T645 rotor (Thermo) at 35,000 rpm for 90 min. Purification by Ni(II) chelate capture and elution with 100 nM of the tag-cleaving bdSENP1 protease was as previously described (*Frey and Görlich, 2014b*). One hundred micrograms of each antigen (in phosphate-buffered saline [PBS]) were used per immunisation with 200 µl Fama as an adjuvant (Gerbu #3030), following two pre-immunisations with myofibrils isolated from flight muscles of 500 adult flies.

Blood sampling, lymphocyte isolation, and construction of an M13 phage display library were done as described previously (*Pleiner et al., 2015*; *Pleiner et al., 2018*). Phage display itself was performed with 1 nM biotinylated baits immobilised to streptavidin magnetic beads. Selected clones were sequenced in a 96-well format. Coding sequences were cloned for expression into H14-NEDD8 or His14-ScSUMO vectors, with ectopic cysteines at N- and C-termini of the nanobody. The here described nanobody expression constructs are listed in the Material availability statement at the end of the Methods section and are available at Addgene (https://www.addgene.org/Dirk_Gorlich/).

## Nanobody expression, purification, and labelling

Nanobodies were expressed in NEB Shuffle Express, which allows the structural disulphide bond to be (partially) formed. Bacteria were grown initially in 5-l flasks containing 250 ml TB medium

supplemented with 50 µg/ml kanamycin and 0.5% glucose overnight at 37 °C to stationary phase (OD$_{600}$ ~10). The cultures were then shifted to 21 °C, diluted with 500 ml fresh medium, and induced 20 min later with 100 µM IPTG for 4 hr.

Bacteria were pelleted and resuspended in 50 ml sonication buffer (50 mM Tris/HCl pH 7.5, 20 mM imidazole/HCl pH 7.5, 300 mM NaCl, 5 mM reduced glutathione [GSH], 2.5 mM oxidised glutathione [GSSG]). Lysis was done by one freeze-thaw cycle followed by sonication and ultracentrifugation as described above. The lysates were then either frozen in aliquots and stored at –80 °C until further use or used directly for large-scale purification. For the latter, 30 ml of lysate was bound at 4 °C to 2 ml Ni(II) matrix; the matrix was extensively washed with sonication buffer, followed by protease buffer (50 mM Tris/HCl pH 7.5, 20 mM imidazole/HCl pH 7.5, 300 mM NaCl, 5 mM GSH, 5% w/v glycerol). Elution was done with 50 nM ScUlp1 in protease buffer overnight at 4 °C or for 2 hr at room temperature (RT). Typical yields range between 10 and 50 mg nanobody per litre of culture.

For labelling, we used two different strategies. For in-solution-labelling, we reduced prepurified nanobodies for 5 min with 20 mM dithiothreitol (DTT) on ice. Then, free DTT was removed by gelfiltration on a Nap5 Sephadex G25 column (Cytiva) equilibrated and degassed in 50 mM potassium phosphate pH 6.8, 300 mM NaCl, and 1 mM imidazole (using a sample volume not exceeding 400 µl). Fluorophore-maleimides were dissolved to 10 mM in dimethylformamide, used in ~50% excess over cysteines to be labelled and pipetted into Eppendorf tubes (placed on ice) before the reduced nanobodies were added. The labelling reaction is fast and typically completed within a few minutes. Free fluorophore was then removed by gel filtration on a Nap5 column, equilibrated in 50 mM Tris/HCl pH 7.5, 300 mM NaCl, 10% glycerol (for nanobodies with a negative net charge), or with 100 mM potassium phosphate pH 6.8, 10% glycerol (for nanobodies with a positive net charge). For storage at 4 °C, 0.05% sodium azide was added. Long-term storage was at –80 °C.

Quality control was done by SDS-PAGE. For most fluorophores, unlabelled, single, and double labelled nanobodies are well resolved, which allows for assessing the completeness of the labelling reaction (see *Figure 2A*). Fluorescence images were acquired from unstained/unfixed gels with a Fuji FLA-9000 system. Concentrations of nanobody, fluorophore, and density of labelling were measured photometrically at 280 nm and at the absorption maximum of the used fluorophore. Extinction coefficient of the nanobody at 280 nm was deduced from its amino acid composition and used to calculate the protein concentration, also considering the cross-absorbance of the fluorophore at 280 nm. Extinction coefficients of the fluorophores at 280 nm and the absorption maximum were taken from the respective suppliers.

Alternatively, nanobodies were labelled while bound as His$_{14}$-ScSUMO or His$_{14}$NEDD8 fusions to a Ni(II) chelate matrix. The matrix should be resistant to reduction by DTT. We used here a homemade matrix (*Goerlich and Frey, 2015*); however, the cOmplete His-Tag purification matrix from Roche was working equally well. In brief, 30 µl Ni beads were slightly overloaded with nanobody, typically by binding 650 µl lysate to them (this usually requires titration). The beads were then washed three times in 650 µl sonication buffer; the ectopic cysteines were reduced by a 5-min incubation at 0 °C with 20 mM DTT, 50 mM Tris/HCl pH 7.5, 300 mM NaCl, and 15 mM imidazole pH 7.5. The beads were then washed twice with degassed prelabelling buffer (50 mM potassium phosphate pH 6.8, 15 mM imidazole/HCl pH 7.0, 300 mM NaCl). Two hundred microlitres of labelling solution (100–200 µM fluorophore in 50 mM potassium phosphate pH 6.8, 1 mM imidazole/HCl pH 7.0, 300 mM NaCl) was added; the beads were shaken for 20 min at 0–4 °C, washed twice in prelabelling buffer, once in cleavage buffer (50 mM Tris/HCl pH 7.5, 500 mM NaCl, 20% glycerol), and finally eluted with 100 µl 50 nM ScUlp1 (in cleavage buffer) overnight at 4 °C. The eluates typically contained 100 µM labelled nanobody and 50 nM was typically used for stainings.

## Biolayer interferometry (BLI)

BLI experiments were performed using High Precision Streptavidin biosensors and an Octet RED96e instrument (ForteBio/Sartorius) at 25 °C with PBS pH 7.4, 0.02% (w/v) Tween-20 and 0.1% (w/v) bovine serum albumin as assay buffer. Sls-Nano2, modified via one N-terminal and one C-terminal ectopic cysteine with two Biotin-PEG$_3$-Maleimide molecules (Iris Biotech), was bound at 0.6 µg/ml concentration to the sensors until a wavelength shift/binding signal of 0.4 nm was reached. After one washing step in buffer, the biosensors were dipped into wells containing a concentration series of the Sls-Ig13/14 domains to measure the association rate and then incubated with assay buffer for dissociation.

Data were reference-subtracted, and curves were fitted globally with a 1:1 binding model (Octet Data Analysis HT 12.0 software).

## Myofibril isolation for immunisation

We hand-dissected indirect flight muscles from 1000 adult wild-type flies from the Luminy strain (*Leonte et al., 2021*) in two batches of 500 each. To dissect, we cut away wings, head, and abdomen and separated the thoraces into two halves along the midline using small dissection scissors (#15009–08 Fine Science Tools) and placed them into relaxing solution (100 mM NaCl, 20 mM $NaP_i$ pH 7.2, 6 mM $MgCl_2$, 5 mM ATP, 0.5% Triton X-100, complete protease inhibitor cocktail [Merck, Sigma #11697498001]) with 50% glycerol for a few minutes under the dissection scope. We then cut and scooped out the flight muscles, without taking gut or jump muscles using scissors and fine forceps (#11252–20 Dumont#5, Fine Science Tools). We collected flight muscles from 500 flies in one tube in relaxing buffer plus 50% glycerol and left them up to 24 hr at –20 °C. Then, we spun the myofibrils down at 200 g and washed the pellet with relaxing buffer without glycerol. The purified myofibrils were then frozen in liquid nitrogen and stored at –80 °C until used for alpaca immunisation.

## Fly strains and genetics

Fly stocks were maintained under standard culture conditions (*Avellaneda et al., 2021*). All crosses were developed at 27 °C to enhance RNAi efficiency (*Schnorrer et al., 2010*). Wild-type control flies were *w[1118]*, *Luminy*, or *Mef2*-GAL4 driver crossed to *w[1118]*. To knock-down *sls*, Projectin (*bt*), *Actn*, and *Zasp52* muscle-specific *Mef2*-GAL4 was crossed with *UAS-sls-IR* (TF47301), *UAS-bt-IR* (TF46252), or *UAS-Actn-IR* (TF7760) long ds-RNAi lines obtained from the VDRC stock centre (*Dietzl et al., 2007*) or *UAS-Zasp52-IR* (JF01133) obtained from the Bloomington stock centre (*Ni et al., 2011*) and muscles were stained with nanobodies. *Unc-89[EY15484]* (Obscurin mutant) was obtained also from the Bloomington stock centre (*Katzemich et al., 2012*).

## Embryo fixation and staining

To investigate the larval musculature morphology at embryonic stages 16 and 17, crosses of the correct genotypes were set up in fly cages, in the presence of apple juice agar plates and a drop of yeast paste at 27 °C. Flies were allowed to lay overnight, and the next day, the embryos were collected and aged for at least another 8 hr at 27 °C. For fixation, embryos were dechorionated in 50% bleach for 2–3 min and then fixed for 20 min with a 1:1 mixture of 4% paraformaldehyde (PFA in fresh PBS) and heptane in glass tubes on a shaker at RT. To free the embryos from the vitelline membrane, the fixative (lower phase) was removed with a glass pipette, one volume of methanol (MeOH) was added, and the tube was shaken vigorously. Dechorionated embryos sank to the bottom and were washed 3× with MeOH. Embryos were stored at –20 °C in MeOH.

For antibody and nanobody stainings, embryos were rehydrated in PBS-T (PBS with 0.3% Triton-X-100), blocked for more than 30 min with 4% normal goat serum and stained with fluorescently labelled nanobodies alone, or together with antibodies, overnight in PBS-T. Antibodies were visualised with standard secondary antibodies (Molecular Probes, 1/500 in PBS-T), and embryos were mounted in SlowFadeTM Gold Antifade (Thermo Fisher), and imaged with a Zeiss LSM880 confocal microscope using 40× or 63× objectives.

## Flight and leg muscle staining

Flight and leg muscles were stained, as previously described in detail (*Weitkunat and Schnorrer, 2014*). Briefly, wings, head, and abdomen were clipped from adult flies with fine scissors, and thoraces were fixed in 4% PFA in PBS-T for 20 min at RT. After washing once with PBS-T, the thoraces were placed on a slide with double-sticky tape with the head position facing the sticky tape and cut sagittally with a microtome blade (Pfm Medical Feather C35). Hemithoraces were stained with fluorescent nanobodies and rhodamine-phalloidin (1:1000 Molecular Probes) for 2 hr at RT or overnight at 4 °C. Hemithoraces were washed twice with PBS-T, mounted in SlowFadeTM Gold Antifade (Thermo Fisher) using two coverslips as spacers, and flight or leg muscles were imaged with a Zeiss LSM880 confocal microscope using a 63× objective.

## Analysing antibody versus nanobody labelling intensity decay over depth

We manually drew selections with Fiji (*Schindelin et al., 2012*) on stacks obtained with confocal imaging; each selection consisted of one myofibril. We used these selections to extract intensity profiles that were then analysed automatically using Python custom codes. The automated analysis to extract the intensity of each band consisted of the following: (a) locate bands in profiles using the peak finding algorithm find_peaks from the Scipy library; (b) subtract background on the profile, linear fitting the 35% lowest values of the profile and subtracting this fit on the profile; (c) fit bands on the background-corrected profile with Gaussian functions; and (d) estimate the area under the curve of these fits. This initial analysis allowed us to estimate the integrated intensity of bands of Obscurin-GFP and epitopes labelled with the Kettin (Sls-Ig16) antibody and Sls-Nano2 (Sls-Ig13/14) nanobody. To estimate how fast intensity decays with depth in the confocal z-stacks, for each animal, we fitted an exponential decay function to the averaged band intensity over each selection (a myofibril) versus the depth where it was imaged (*Figure 6—figure supplement 1*). The decay lengths obtained were then reported in *Figure 6C*. In our imaging conditions, the decay of intensity with the depth of GFP was higher than the one of Sls-Nano2, likely caused by faster bleaching of GFP compared to the Alexa488 dye when acquiring a z-stack.

## Dissection and staining of larval muscles

To perform antibody or nanobody stainings of larval muscles, L3 larvae were collected with a brush and placed at 4 °C. For dissection, larvae were covered with HL3 buffer and pinned individually by pushing one insect pin through the head and one through the abdomen to immobilize them in dissection dishes placed on ice (*Stewart et al., 1994*). Pinned larvae were dissected with sharp scissors from the dorsal side in HL3 buffer, and interior organs (gut and fat body) were removed with forceps. The remaining larval fillets were fixed in 4% PFA in PBS-T for 30 min and then blocked in 4% normal goat serum for 30 min at RT on a shaker. Nanobodies and antibodies were incubated in PBS-T overnight at 4 °C. Larval fillets were then washed three times for 10 min in PBS-T at RT and stained with secondary antibodies and phalloidin (labelled with rhodamine 1:1000, Molecular Probes) in PBS-T for 2 hr at RT in the dark. After washing three times with PBS-T for 5 min, larval fillets were mounted in SlowFadeTM Gold Antifade (Thermo Fisher) and imaged with a Zeiss LSM880 confocal microscope using 20×, 40×, or 63× objectives.

To quantify larval sarcomere and Sls length, the images were processed with a Gaussian blur (sigma: 1.00) and a line perpendicular to the Z-disc was drawn to retrieve an intensity profile. The position of the peak of intensity was determined by using the BAR plugin in Fiji (*Schindelin et al., 2012*). Sarcomere length was calculated by the distance between two peaks of Sls-Nano2 staining, and Sls length by the distance between a peak of Sls-Nano2 and one of Sls-Nano42.

## Quantification of staining intensities in deGrad experiments

To efficiently quantify the effect of the deGrad system on nanobodies staining intensities in sarcomeres, we created a Fiji macro toolset. First, confocal images of larval muscle were corrected for background: for this purpose, we selected an area of about $10 \times 10$ μm in the image without myofibrils and estimated the average pixel intensity. This value was then subtracted from the image. Second, to estimate the relative amount of nanobodies, we positioned rectangular selections encompassing regions of sarcomeres labelled by nanobodies, from which we extracted the average intensity and repeated this to have 100 selections. The rectangular selections had the same size in all measurements to ensure reproducibility.

## Generation of *UAS-Nano-mNeonGreen* transgenic flies

To clone *UAS-Sls-Nano2-NeonGreen*, we linearised pUAST-attB with EcoRI and inserted mNeonGreen by Gibson Assembly (Gibson Assembly) after amplification of mNeonGreen with 5′-ACTCTGAATAGG GAATTGGGAATTC-3′ and 5′-CGGCCGCAGATCTGTTAAC-3′ primers. In a second step, we linearised pUAST-attB-mNeonGreen with EcoRI and inserted the Sls-Nano2 sequence by Gibson Assembly (Gibson Assembly) after amplification with 5′- ACTCTGAATAGGGAATTGGG-3′ and 5′-CCTTGCTC ACCATGGAAC-3′ primers. For transgenesis, we injected the pUAST-attB-Sls-Nano2-mNeonGreen

plasmid into the attP landing site strain VK00033 located at 65B on the third chromosome by standard injection and selection methods (*Sarov et al., 2016*).

To clone *UAS-Sls-Nano42-NeonGreen*, *UAS-Proj-Nano30-NeonGreen*, and *UAS-Proj-Nano37-NeonGreen,* we amplified nanobodies with primers 5′-TTTGAATTCCCCGCCATGGGCCAGGTGCAATTGGTAGA-3′ and 5′-AAAAGCGGCCGCACATGACGTTGATGAGACTGTGAC-3′. After enzymatic digestion of the amplification products and linearisation of a modified pUAST-attB-mNeonGreen with NotI and EcoRI, the amplification products were cloned into pUAST-attB-mNeonGreen, and clones were injected into attP site VK00033 by standard methods (*Sarov et al., 2016*).

## Generation of *UAS-sls-Nano-deGrad flies*

To create *UAS-sls-Nano-deGrad flies*, we fused the F-box domain contained in the N-terminal part of Slmb (NSlmb) to either Sls-Nano2 or Sls-Nano42. We amplified NSlmb from the *Drosophila* line *UAS-NSlmb-vhhGFP4* (*Caussinus et al., 2011*) using primers 5′-GGGGGAATTCAAAATGATGAAAATGG-3′ and 5′-CCATCTCGAGGTGGCGGCCAG-3′, *sls-Nano2* with primers 5′-TTTCTCGAGCCCGCCATGGGCCAGGTGCAATTGGTAGA-3′ and 5′-AAAAGCGGCCGCTTATGAGGTACTGGAGACGGTGACCC-3′ and *sls-Nano42* with primers 5′-TTTCTCGAGCCCGCCATGGGCCAGGTGCAATTGGTAGA-3′ and 5′-GGAAGCGGCCGCTTAACATGACGTTGATGAGACTGTGAC-3′. After enzymatic digestion (EcoRI/XhoI and XhoI/NotI), the amplification products were cloned in pUASTattB and injected into the attP landing site VK00033 using standard methods (*Sarov et al., 2016*).

## Larval crawling

L3 larvae were collected at the wandering stage, placed in a 15 cm petri dish filled with 2% agarose, and allowed to acclimatise for at least 20 min at RT. Then, larvae were placed simultaneously in the centre of the dish and imaged at a frame rate of 25 Hz. Images were acquired using an infrared Basler acA2040-90 µm NIR camera equipped with a Kowa LM12SC lens and a homemade LED infrared illumination system (WINGER WEPIR3-S1 IR Power LED Star infrared at 850 nm). The Pylon viewer software from Basler was used to control acquisition, and exposure time was adjusted for enhanced contrast. The assays were repeated at least two times for each genotype, with assays done on different days. The videos were analysed using FIMTrack (*Risse et al., 2017*), and data were visualized via Python.

## Live imaging of larval muscles

To quantify Sls-Nano2 localisation *in vivo*, we crossed *UAS-Sls-Nano2-NeonGreen* flies with *Mef2-GAL4* and collected L3 larvae. To reduce the movement of the living larvae, larvae were anaesthetised for 5 min with diethyl ether (Aldrich) (*Kakanj et al., 2020*) and then mounted in 10 S halocarbon oil. Larvae were imaged with an Olympus spinning disc confocal microscope with a 60 × objective. Photobleaching was performed with a 488 nm laser (Rapp-opto), and recovery was quantified for 30 min. Regions of interest (20 × 10 µm) inside the bleached area, in the non-bleached area, or outside the muscle as background were selected, and their intensities were measured at each time point. To calculate the ratio of FRAP, the intensity of the bleached area background subtracted was divided by the intensity in the non-bleached area background subtracted.

## Materials availability statement

Newly generated code is publicly available here: https://github.com/PierreMangeol/titin_PAINT (*Loreau, 2022* copy archived at swh:1:rev:95e2ac29f658f8fca2435d93ab3c6326c786047d) *E. coli* nanobody expression vectors are available from Addgene (https://www.addgene.org/Dirk_Gorlich/). Requestees are asked to quote the appropriate plasmid numbers:

| Plasmid number | Nanobody name | Nanobody clone ID | Target | Expressed protein | AddGene ID |
|---|---|---|---|---|---|
| pDG03139 | Sls-Nano2 | NbRe11 | Sls Ig13-14 | H14-NEDD8-Nb-Cys | Addgene_195990 |
| pDG03248 | Sls-Nano2 | NbRe11 | Sls Ig13-14 | H14-SUMO-Cys-Nb-Cys | Addgene_195991 |
| pDG03776 | Sls-Nano39 | Re1F04 | Sls Ig49/50 | H14-NEDD8-Nb-Cys | Addgene_195992 |
| pDG03247 | Sls-Nano42 | Re1G12 | Sls Ig51-Fn2 | H14-SUMO-Cys-Nb-Cys | Addgene_195993 |

*Continued on next page*

*Continued*

| Plasmid number | Nanobody name | Nanobody clone ID | Target | Expressed protein | AddGene ID |
|---|---|---|---|---|---|
| pDG03777 | Sls-Nano42 | Re1G12 | Sls Ig51-Fn2 | H14-NEDD8-Nb-Cys | Addgene_195994 |
| pDG03781 | Sls-Nano48 | Re1G01 | Sls Ig51-Fn2 | H14-NEDD8-Nb-Cys | Addgene_195995 |
| pDG03769 | Proj-Nano28 | Re1A01 | Proj Fn1/2 | H14-NEDD8-Nb-Cys | Addgene_195996 |
| pDG03246 | Proj-Nano29 | Re1A02 | Proj Ig5-8 | H14-SUMO-Cys-Nb-Cys | Addgene_195997 |
| pDG03770 | Proj-Nano29 | Re1A02 | Proj Ig5-8 | H14-NEDD8-Nb-Cys | Addgene_195998 |
| pDG03771 | Proj-Nano30 | Re1B01 | Proj Ig5-8 | H14-NEDD8-Nb-Cys | Addgene_195999 |
| pDG03772 | Proj-Nano33 | Re1C06 | Proj Ig27-Fn35 | H14-NEDD8-Nb-Cys | Addgene_196000 |
| pDG03773 | Proj-Nano34 | Re1D11 | Proj kinase | H14-NEDD8-Nb-Cys | Addgene_196001 |
| pDG03774 | Proj-Nano35 | Re1E11 | Proj kinase | H14-NEDD8-Nb-Cys | Addgene_196002 |
| pDG03775 | Proj-Nano37 | Re1E12 | Proj kinase | H14-NEDD8-Nb-Cys | Addgene_196003 |
| pDG03779 | Proj-Nano46 | Re1E10 | Proj kinase | H14-NEDD8-Nb-Cys | Addgene_196004 |
| pDG04093 | Obs-Nano55 | Re33A03 | Obs SH3-RhoGEF | H14-SUMO-Cys-Nb-Cys | Addgene_196005 |
| pDG04095 | Obs-Nano56 | Re33F04 | Obs SH3-RhoGEF | H14-SUMO-Cys-Nb-Cys | Addgene_196006 |
| pDG04096 | Obs-Nano57 | Re33H04 | Obs SH3-RhoGEF | H14-SUMO-Cys-Nb-Cys | Addgene_196007 |
| pDG04091 | Obs-Nano58 | Re33D05 | Obs Ig13-17 | H14-SUMO-Cys-Nb-Cys | Addgene_196008 |
| pDG04092 | Obs-Nano59 | Re33F06 | Obs Ig13-17 | H14-SUMO-Cys-Nb-Cys | Addgene_196009 |
| pDG04108 | Actn-Nano62 | Bm15B04 | Actn CH1-Spec4 | H14-SUMO-Cys-Nb-Cys | Addgene_196010 |
| pDG04109 | Actn-Nano63 | Bm15D02 | Actn CH1-Spec4 | H14-SUMO-Cys-Nb-Cys | Addgene_196011 |
| pDG04110 | Actn-Nano64 | Bm15D04 | Actn CH1-Spec4 | H14-SUMO-Cys-Nb-Cys | Addgene_196012 |
| pDG04135 | Zasp-Nano65 | Re38E02 | Zasp52 PDZ | H14-SUMO-Cys-Nb-Cys | Addgene_196013 |
| pDG04136 | Zasp-Nano66 | Re38F05 | Zasp52 PDZ | H14-SUMO-Cys-Nb-Cys | Addgene_196014 |

## Acknowledgements

We thank Stefan Raunser and Mathias Gautel and all their group members, as well as the Schnorrer and Görlich groups for their stimulating discussions within the StuDySARCOMERE ERC synergy grant. We thank Metin Aksu for help with the Octet measurements. We thank Ulrike Teichmann and her team for alpaca care and immunisations. We are grateful to the IBDM imaging and fly facilities for help with image acquisition, maintenance of the microscopes, and fly food. This work was supported by the Centre National de la Recherche Scientifique (CNRS, FS, CP, NML), the Max Planck Society (DG), Aix-Marseille University (PM), the European Research Council under the European Union's Horizon 2020 Programme (ERC-2019-SyG 856118 to DG and FS), the excellence initiative Aix-Marseille University A*MIDEX (ANR-11-IDEX-0001–02, FS), the French National Research Agency with ANR-ACHN MUSCLE-FORCES (FS), the Human Frontier Science Program (HFSP, RGP0052/2018, FS), the Bettencourt Schueller Foundation (FS), the France-BioImaging national research infrastructure (ANR-10-INBS-04–01) and by funding from France 2030, the French Government program managed by the French National Research Agency (ANR-16-CONV-0001) and from Excellence Initiative of Aix-Marseille University - A*MIDEX (Turing Center for Living Systems) and LabEx-INFORM (FS and VL). The funders had no role in study design, data collection and analysis, decision to publish, or preparation of the manuscript.

## Additional information

### Funding

| Funder | Grant reference number | Author |
|---|---|---|
| Centre National de la Recherche Scientifique | | Frank Schnorrer<br>Nuno Miguel Luis<br>Christophe Pitaval |
| Max-Planck-Gesellschaft | | Dirk Görlich |
| Aix-Marseille Université | | Pierre Mangeol |
| European Research Council | ERC-2019-SyG 856118 | Dirk Görlich<br>Frank Schnorrer |
| Aix-Marseille Université | A*MIDEX | Frank Schnorrer |
| Agence Nationale de la Recherche | ANR-11-IDEX-0001-02 | Frank Schnorrer |
| Agence Nationale de la Recherche | ANR-ACHN MUSCLE-FORCES | Frank Schnorrer |
| Human Frontier Science Program | RGP0052/2018 | Frank Schnorrer |
| Bettencourt Schueller Foundation | | Frank Schnorrer |
| Agence Nationale de la Recherche | ANR-10-INBS-04-01 | Frank Schnorrer |
| Agence Nationale de la Recherche | ANR-16-CONV-0001 | Frank Schnorrer |
| Aix-Marseille Université | Center for Living Systems | Frank Schnorrer |
| Aix-Marseille Université | LabEx-INFORM | Vincent Loreau |

The funders had no role in study design, data collection and interpretation, or the decision to submit the work for publication.

### Author contributions

Vincent Loreau, Data curation, Formal analysis, Investigation, Visualization, Methodology, Writing - original draft, Writing - review and editing; Renate Rees, Eunice HoYee Chan, Waltraud Taxer, Data curation, Formal analysis, Investigation; Kathrin Gregor, Bianka Mußil, Data curation, Investigation; Christophe Pitaval, Data curation, Methodology; Nuno Miguel Luis, Data curation, Investigation, Methodology, Writing - review and editing; Pierre Mangeol, Software, Formal analysis, Investigation, Visualization, Methodology, Writing - review and editing; Frank Schnorrer, Conceptualization, Formal analysis, Supervision, Funding acquisition, Investigation, Visualization, Methodology, Writing - original draft, Writing - review and editing; Dirk Görlich, Conceptualization, Supervision, Funding acquisition, Methodology, Writing - original draft, Writing - review and editing

### Author ORCIDs

Vincent Loreau (iD) http://orcid.org/0000-0002-0556-2825
Eunice HoYee Chan (iD) http://orcid.org/0000-0003-3162-3609
Nuno Miguel Luis (iD) http://orcid.org/0000-0001-5438-9638
Pierre Mangeol (iD) http://orcid.org/0000-0002-8305-7322
Frank Schnorrer (iD) http://orcid.org/0000-0002-9518-7263
Dirk Görlich (iD) http://orcid.org/0000-0002-4343-5210

### Decision letter and Author response

Decision letter https://doi.org/10.7554/eLife.79343.sa1
Author response https://doi.org/10.7554/eLife.79343.sa2

# Additional files

## Supplementary files
• MDAR checklist

## Data availability
All quantitative source data are provided. Newly generated code is publicly available here: https://github.com/PierreMangeol/titin_PAINT (copy archived at swh:1:rev:95e2ac29f658f8fca2435d-93ab3c6326c786047d) *E. coli* nanobody expression vectors are available from Addgene (https://www.addgene.org/Dirk_Gorlich/).

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
