## [Editor Report]

In this important study, the authors have generated a large collection of nanobodies against *Drosophila* muscle components, showing their interest to define the molecular organisation of muscle sarcomeres. Moreover, they show that nanobody expression in muscles can block the normal function of those proteins. While the use of nanobodies to reveal the distribution of proteins as such is not novel, their use in a model organism is novel and their demonstration of their usefulness is compelling. Beyond *Drosophila* and muscles their work can emulate similar strategies for other tissues in other species.

---

## [Decision Letter]

**Decision letter after peer review:**

Thank you for submitting your article "A nanobody toolbox to investigate localisation and dynamics of *Drosophila* titins" for consideration by *eLife*. Your article has been reviewed by 4 peer reviewers, including Michel Labouesse as Reviewing Editor and Reviewer #1, and the evaluation has been overseen by Anna Akhmanova as the Senior Editor. The following individuals involved in the review of your submission have agreed to reveal their identity: Hidde L Ploegh (Reviewer #2); Guy M Benian (Reviewer #3).

Essential evisions:

The three external reviewers for each of your two manuscripts, along with myself as a guest editor, are overall supportive of your work. We think that you are introducing interesting tools, although one of the reviewers thinks that you overemphasized the novelty of nanobodies as an approach. The model you propose for sarcomere localization of the titin-like fly homologs Sallimus and Projectin is interesting and should inspire further work.

That being said, after exchanging views among the reviewers, we feel that there is probably too much redundancy between the two manuscripts in terms of methods and overall conclusions.

1) We suggest that you merge both manuscripts. This would allow you to show that your data are consistent among the two different muscle types (leg muscle and flight muscle), with an overlap in localization of the C-term of Sls with the N-term of Projectin at the outer edge of the A-band. The generation of the nanobodies and their ability to penetrate tissues would be described only once.

2) In terms of additional experiments, we ask that you use a nanobody-GFP to determine whether the expression of the nanobody fused to mNeonGreen in muscle might affect muscle function or affect the behavior of the fusion. You could easily test this by performing some sort of motility assays on the larva and determine whether muscle sarcomere structure is affected after immunostaining with various marker antibodies. Alternatively, you could use the nanobody-GFP to follow the course of a contraction/relaxation cycle in vivo. When reorganizing your manuscripts into a single one, please give better credit to the *C. elegans* muscle field (as requested by one reviewer), and generally attend to other comments formulated by the reviewers.

*Reviewer #1 (Recommendations for the authors):*

Comments:

Please further prove that your nanobodies can be used in functional tools, along the lines that you outlined in the discussion.

*Reviewer #2 (Recommendations for the authors):*

The discussion mentions a number of possible nanobody applications (lines 347-354), none of which are pursued in this manuscript. References to reviews (of which there are quite a few) that describe such applications would therefore be sufficient.

I found the writing to be rather imprecise/sloppy and ambiguous in places. Excess verbiage such as non-essential adjectives and adverbs should be removed wherever possible. In general, the overuse of gerunds is to be discouraged, as is the repeated use of 'toolbox'. The logical conjunction 'and' should not be used to string disparate clauses together.

To avoid lab jargon, the correct descriptor of nanobodies should include the prefix 'anti'. Therefore: anti-Projectin nanobodies' instead of "Projectin nanobodies'. The authors should decide on using either 'N-term 'or 'N-terminus', 'C-term' or 'C-terminus' and be consistent throughout the manuscript. I don't think the term 'N-terminal nanobody' or 'C-terminal nanobody' can be used as shorthand for a 'nanobody that recognizes the N- or C-terminal segment of …' I have corrected this for a number of occurrences but gave up after a while.

*Reviewer #3 (Recommendations for the authors):*

I have only a few suggestions for this really excellent, timely, and innovative study and report. First, there should be more rationale given for their approach to generating the nanobodies. For example, the authors should state why Alpacas vs. some other animal species like rabbit or goat were chosen, and more importantly, why did they first immunize with flight muscle myofibrils before immunizing with the various recombinantly expressed domains? What was the reason for not just simply immunizing with the domains? Second, the authors mention several times that their staining of various muscle types confirmed the existence of different Sallimus and Projectin isoforms in different muscle types. Could they confirm this idea by performing western blots with at least some of their nanobodies? I know that doing western blots for such large polypeptides is challenging, but it can be done, especially with lower percentage acrylamide gels or even agarose gels as has been done for mammalian titin. However, I would not require this data in the current manuscript. Just saying that it would be nice to see at least some of this, and would also demonstrate the utility of nanobodies for western detection; indeed with their higher affinities, perhaps detection levels are also enhanced, compared to use of conventional antibodies. Third, in addition to my comment in the Public Review about the authors ignoring work in other invertebrates especially *C. elegans*, in the Discussion, the authors make an interesting comment regarding Projectin, "Hence, it will be interesting to resolve in the future if the kinase activity is required for sarcomere formation or function." At the risk of tooting my lab's own work, we have already investigated this question for twitchin (Projectin) in *C. elegans*. In Matsunaga et al. (2017), we used CRISPR to gene edit the twitchin gene so that it expressed a full-length abundant twitchin in which the protein kinase domain had been rendered catalytically inactive. These worms have normal sarcomere structure but move faster and contract more than wild type worms, suggesting that the normal function of the catalytic activity of twitchin kinase is to inhibit muscle activity.

*Reviewer #4 (Recommendations for the authors):*

Authors test the specificities of their Nanobodies in late embryos using muscle-specific RNAi attenuation of sls and bent. Among them, Sls-Nano2 is expected to detect both short and long Sls isoforms whereas Sls-Nano39 and Sls-Nano42 only the long ones. Staining of all of them is strongly reduced in Sls-RNAi context. What could be interesting to test is:

i) Whether nanobodies directed to the same protein region (here Sls C-term rich of Ig repeats) have similar efficacy and specificity of detection.

ii) Whether the expression of short Sls isoform detected by the Sls-Nano2 and potentially not overlapping with Projectin could be specific for developmental stages before the sarcomere assembly.

One potential application of Nanobodies is their use in following native proteins in vivo. Because proteins tagged with GFP are often functionally affected it would be of interest to test whether the functionality of proteins to which Nonobodies bind is preserved. Here authors could test whether Sls-Nano2NeoGreen expressed in muscle has an impact or not on Sls muscle function.

---

## [Author Response]

Essential revisions:The three external reviewers for each of your two manuscripts, along with myself as a guest editor, are overall supportive of your work. We think that you are introducing interesting tools, although one of the reviewers thinks that you overemphasized the novelty of nanobodies as an approach. The model you propose for sarcomere localization of the titin-like fly homologs Sallimus and Projectin is interesting and should inspire further work.That being said, after exchanging views among the reviewers, we feel that there is probably too much redundancy between the two manuscripts in terms of methods and overall conclusions.

As explained in our initial rebuttal letter, we have the opinion that both papers should stand alone. To substantiate this, we have largely expanded the here presented nanobody toolbox paper. We have added 10 additional new nanobodies against different domains of the key sarcomeric proteins Obscurin, α-Actinin and Zasp52 to make the resource more complete. As α-Actinin and Zasp52 are not muscle specific, we expect these nanobodies to be used by the wide community of *Drosophila* epithelial biologists. We have further included the detailed characterisations of the Sls-Nano2-NeonGreen expressing larval muscles and have generated additional Nanobody-NeonGreen fusion fly strains. Finally, we provide a first analysis of two Sls nanobodies tagged with a deGrad degradation signal in vivo. This shows an interesting modification of parts of the Sls protein in vivo.

Furthermore, to remove redundancy, we have moved the nanobody penetration data in flight muscles from the PAINT paper to this toolbox paper and also moved almost all the confocal analysis to the toolbox paper, as requested.

1) We suggest that you merge both manuscripts. This would allow you to show that your data are consistent among the two different muscle types (leg muscle and flight muscle), with an overlap in localization of the C-term of Sls with the N-term of Projectin at the outer edge of the A-band. The generation of the nanobodies and their ability to penetrate tissues would be described only once.

As already explained in the response letter of the DNA-PAINT manuscript: we have moved almost all the confocal microscopy part, including the quantitative antibody penetration analysis to this toolbox manuscript. We only left a basic Figure 1 in the DNA-PAINT manuscript to be able to understand it for the reader as a stand-alone paper.

We want to briefly reiterate why the nanobody toolbox resource paper should stand on its own:

1. It now reports the generation of the *Drosophila* sarcomere protein nanobody resource (22 different nanobodies against 11 different epitopes) – we would like to remind the reviewers that not a single nanobody against a fly protein domain has been published to date. Thus, our effective pipeline should inspire future work in various directions for the large fly community. The anti-α-Actinin and Zasp52 nanobodies will be applicable to many different tissues, in particular epithelial tissues.

In this paper, we do include all data from all three muscle types, larval, leg and flight muscles, so these data are not spread over the 2 papers (in particular after including most of Figures1 and 2 from the PAINT paper in the revised toolbox papers, thanks for suggesting this).

2. We demonstrate the superior labelling and penetration efficiencies of our nanobodies compared to antibodies in embryonic and flight muscles.

3. The paper shows that nanobodies against fly proteins (and not only against GFP) can be used to live image the dynamics of the target protein. Biologically, this shows that Sallimus (fly I-band titin) once incorporated into the sarcomere is not mobile anymore. This contrasts publications in cultured cardiomyocytes that suggested rather fast mobility of mammalian titin, likely because of the limited sarcomere maturation in vitro (PMID 31757849). Hence, we provide an important finding relevant for sarcomere homeostasis.

We now also show data that larval crawling speed and larval muscle structure is not affected by the nanobody expression, thanks for suggesting these important controls.

4. We further show that both Sallimus and Projectin are needed for their striated localisation as soon as sarcomeres form in late-stage embryos. Although this was assumed before, it was never shown, as Sls or Projection mutants die as embryos and staining of latestate embryos is hardly feasible with traditional antibodies. However, only late stage 17 embryos do show striated muscles. Again, an important finding for the fly muscle field that is relevant to the mammalian field too, as it suggests that only Sallimus together with Projectin can fulfil the mammalian titin function.

5. Using the nanobodies, we found that Sls protein is extending over more than 2 µm in relaxed larval muscle to reach the myosin filament (longer than the 1.5µm of the long human titin isoform!). This is a significant finding as the *Drosophila* titins were always considered too short to act as bona fide ruler proteins. We now cite a paper from crab muscle showing that *Drosophila* is not the exception and thus hopefully our findings will convince the mammalian muscle community that Sls and Projectin are true functional homologs of mammalian titin.

6. Furthermore, using our nanobodies, we found that Projectin is incorporated in a polar orientation into the thick filament. Together, with its essential role to form a striated larval sarcomere this finding is stressing the functional homology of the fly titin homologs with the single mammalian titin homolog once more. It may also open a new way of thinking how thick filament length is controlled outside of vertebrate muscle. This should finally convince the mammalian field that Sls and Projectin are functional titin homologs.

7. Finally, our first proof of principle data on the Sls-nanobody-deGrad fusions open interesting new avenues for future experiments during sarcomere development and sarcomere homeostasis that we are not able to fully extent here in this resource manuscript.

Together, we think these findings are justifying an *eLife* resource paper. In any case, if a back-to-back publication of the two manuscripts in *eLife* is not accepted, then we will move with both to another Journal.

2) In terms of additional experiments, we ask that you use a nanobody-GFP to determine whether the expression of the nanobody fused to mNeonGreen in muscle might affect muscle function or affect the behavior of the fusion. You could easily test this by performing some sort of motility assays on the larva and determine whether muscle sarcomere structure is affected after immunostaining with various marker antibodies. Alternatively, you could use the nanobody-GFP to follow the course of a contraction/relaxation cycle in vivo. When reorganizing your manuscripts into a single one, please give better credit to the *C. elegans* muscle field (as requested by one reviewer), and generally attend to other comments formulated by the reviewers.

Thanks for these valuable suggestions, we have done these experiments and text changes. We detail these changes below.

Reviewer #1 (Recommendations for the authors):Comments:Please further prove that your nanobodies can be used in functional tools, along the lines that you outlined in the discussion.

As explained above we have now included additional nanobodies that will be of general use outside of the muscle community and provide new data showing that the NeonGreen nanobody fusions are valuable tools as well as show data on the nanobody deGrad fusion, which result in interesting loss of function phenotypes.

Reviewer #2 (Recommendations for the authors):The discussion mentions a number of possible nanobody applications (lines 347-354), none of which are pursued in this manuscript. References to reviews (of which there are quite a few) that describe such applications would therefore be sufficient.I found the writing to be rather imprecise/sloppy and ambiguous in places. Excess verbiage such as non-essential adjectives and adverbs should be removed wherever possible. In general, the overuse of gerunds is to be discouraged, as is the repeated use of 'toolbox'. The logical conjunction 'and' should not be used to string disparate clauses together.To avoid lab jargon, the correct descriptor of nanobodies should include the prefix 'anti'. Therefore: anti-Projectin nanobodies' instead of "Projectin nanobodies'. The authors should decide on using either 'N-term 'or 'N-terminus', 'C-term' or 'C-terminus' and be consistent throughout the manuscript. I don't think the term 'N-terminal nanobody' or 'C-terminal nanobody' can be used as shorthand for a 'nanobody that recognizes the N- or C-terminal segment of …' I have corrected this for a number of occurrences but gave up after a while.

We appreciate the energy of this reviewer to change and improve our writing style. We went through all the suggestions and mostly included them. Thanks for this extensive effort. We hope this reviewer will appreciate our revised version.

Reviewer #3 (Recommendations for the authors):I have only a few suggestions for this really excellent, timely, and innovative study and report. First, there should be more rationale given for their approach to generating the nanobodies. For example, the authors should state why Alpacas vs. some other animal species like rabbit or goat were chosen, and more importantly, why did they first immunize with flight muscle myofibrils before immunizing with the various recombinantly expressed domains? What was the reason for not just simply immunizing with the domains?

As we have injected all the domains into the same alpaca. We rationalised that a base immunisation against all the abundant *Drosophila* proteins can be advantageous. We have added a sentence in the Results section.

Second, the authors mention several times that their staining of various muscle types confirmed the existence of different Sallimus and Projectin isoforms in different muscle types. Could they confirm this idea by performing western blots with at least some of their nanobodies? I know that doing western blots for such large polypeptides is challenging, but it can be done, especially with lower percentage acrylamide gels or even agarose gels as has been done for mammalian titin. However, I would not require this data in the current manuscript. Just saying that it would be nice to see at least some of this, and would also demonstrate the utility of nanobodies for western detection; indeed with their higher affinities, perhaps detection levels are also enhanced, compared to use of conventional antibodies.

We have not tried our nanobodies for western blotting. However, it established that typical nanobodies recognise folded domains and do not work with SDS-denatured samples. Exceptions from this rule are very rare, and with our choice of immunogens it is even more unlikely that linear (Western blot compatible) epitopes recognised. In any case, the presence of different Sls isoforms is already documented in the literature on RNA (Spletter et al. 2018) and protein level (Lakey et al. 1997; Burkart et al. 2007), we cite these papers.

Third, in addition to my comment in the Public Review about the authors ignoring work in other invertebrates especially *C. elegans*, in the Discussion, the authors make an interesting comment regarding Projectin, "Hence, it will be interesting to resolve in the future if the kinase activity is required for sarcomere formation or function." At the risk of tooting my lab's own work, we have already investigated this question for twitchin (Projectin) in C. elegans. In Matsunaga et al. (2017), we used CRISPR to gene edit the twitchin gene so that it expressed a full-length abundant twitchin in which the protein kinase domain had been rendered catalytically inactive. These worms have normal sarcomere structure but move faster and contract more than wild type worms, suggesting that the normal function of the catalytic activity of twitchin kinase is to inhibit muscle activity.

Thanks for highlighting the really interesting paper that we had missed. We now included it, as well as the other *C. elegans* paper, in our discussion.

We again want to thank this reviewer for appreciating our manuscript and help making it more widely relevant.

Reviewer #4 (Recommendations for the authors):Authors test the specificities of their Nanobodies in late embryos using muscle-specific RNAi attenuation of sls and bent. Among them, Sls-Nano2 is expected to detect both short and long Sls isoforms whereas Sls-Nano39 and Sls-Nano42 only the long ones. Staining of all of them is strongly reduced in Sls-RNAi context. What could be interesting to test is:i) Whether nanobodies directed to the same protein region (here Sls C-term rich of Ig repeats) have similar efficacy and specificity of detection.

Is the reviewer asking is Sls-Nano42 and Sls-Nano48 (both against SlsIg51-Fn2) bind with the same affinity? We have not tested this in detail. In immunostainings they show comparable intensities, similar to Sls-Nano39 (against Sls-Ig49/50). See Figure 5 and Figure 5 Supplement.

ii) Whether the expression of short Sls isoform detected by the Sls-Nano2 and potentially not overlapping with Projectin could be specific for developmental stages before the sarcomere assembly.

In the embryo, at stage 16 before the first sarcomeres assemble, both long and short isoforms are likely present. At least the long isoform is present as Sls-Nano2, Sls-Nano39 and Sls-Nano42 give a good signal in stainings (Figure 4). We believe that without the long Sls isoform actin cannot be stably linked to myosin, hence a sarcomere cannot be assembled.

One potential application of Nanobodies is their use in following native proteins in vivo. Because proteins tagged with GFP are often functionally affected it would be of interest to test whether the functionality of proteins to which Nonobodies bind is preserved. Here authors could test whether Sls-Nano2NeoGreen expressed in muscle has an impact or not on Sls muscle function.

We thank the reviewer for this valuable comment. We have now included detailed analysis of larval crawling and sarcomere morphology in the Sls-Nano2-NeonGreen expressing muscle and find that they behave normally.